



**Digital map of the Coral Triangle: An online atlas for marine biodiversity conservation**

Irawan Asaad[1,2*], Carolyn J. Lundquist[1,3], Mark V. Erdmann[4], Mark J. Costello[1]

[1] Institute of Marine Science, University of Auckland, Auckland, New Zealand
[2] Ministry of Environment and Forestry, Jakarta, Indonesia
[3] National Institute of Water & Atmospheric Research, Hamilton, New Zealand
[4] Conservation International-Asia Pacific Marine Programs, Auckland, New Zealand

**Corresponding Author:** Irawan Asaad

**Email address**: i.asaad@auckland.ac.nz.



## 1.  Abstract

An online atlas of the Coral Triangle region of the Indo Pacific biogeographic realm was developed. This online atlas consists of the three interlinked digital maps: (1) Biodiversity Features; (2) Areas of Importance for Biodiversity Conservation; (3) Recommended Priorities for Marine Protected Area (MPA) Network Expansion (*www.marine.auckland.ac.nz/CTMAPS*). The first map, Biodiversity Features, provides comprehensive data on the region's marine protected areas and biodiversity features, threats and environmental characteristics. The second provides spatial information on areas of high biodiversity conservation values, while the third map shows priority areas for expanding the current Coral Triangle MPA network. This digital map provides the most comprehensive biodiversity datasets yet assembled for the region. The datasets were retrieved and generated systematically from various open-access sources. To engage a wider audience and to raise participation in biodiversity conservation, the maps were designed as an interactive and online atlas. This digital map presents representative information to promote a better understanding of the key marine and coastal biodiversity characteristics of the region and enables the application of marine biodiversity informatics to support marine ecosystem-based management in the Coral Triangle region.

## 2.  Introduction

The advancement of internet technology has led to the development of marine biodiversity informatics, namely information technologies that are employed to support the management of data and information on marine biodiversity (Bisby, 2000; Heidorn, 2011; Parr & Thessen, 2018). They enable people to freely access primary and secondary data over online-systems, promote integration of data across datasets, and facilitate collaboration between parties (Costello & Vanden Berghe, 2006). Publicly available biodiversity information is important for engaging the public and policymakers to address global issues that threaten ecosystem services and functions such as biodiversity loss, climate change, habitat destruction and overfishing (Costello, 2009a). Integration of data across disciplines is increasingly imperative, as biodiversity research requires interactions with other related fields (*e.g.,* genomics, oceanography, climatology, evolution) to foster better analyses and interpretations (Reichman *et al.*, 2011; Costello *et al.*, 2013).

There has been massive improvement in online biodiversity databases covering species names (*e.g.,* WoRMS (Horton *et al.*, 2016)), species occurrence records (*e.g.,* GBIF (www.gbif.org), OBIS (OBIS, 2015)), species ranges ((*e.g.,* Map of Life (mol.org), IUCN Redlist (www.iucnredlist.org), AquaMaps (www.aquampas.org)), and taxa specific information (*e.g.,* FishBase (Froese and Pauly, 2000), AlgaeBase (Guiry, 2018), sea turtles (Kot *et al.*, 2015)), that are managed, curated and supported by international projects and initiatives. However, the culture of data publishing is still a concern (Costello *et al.,* 2013). Less than 1% of ecological data is accessible after publication (Reichman *et al.*, 2011) and more than 57% of the papers in environmental biology publications examined in a 2011 review had not released their data (Alsheikh-Ali *et al.*, 2011).

Biodiversity Informatics is expected to grow exponentially. Software, infrastructure, and management tools to store, publish and share biodiversity data, particularly over the internet and World Wide Web, have been improved significantly in recent years (Michener, 2015). Such development is supported by the availability of metadata standards to facilitate description of datasets



and data records (*e.g.,* Ecological Metadata Language (EML) (Michener *et al.*, 1997); Darwin core
(GBIF, 2010)), widely-assessed repositories to deposit ecologically-relevant data (*e.g.,* Drayd
(datadryad.org); Figshare (figshare.com); KNB (knb.ecoinformatics.org)) and a variety of open source
data management tools (*e.g., MySQL, R*, and *Kepler*).
Geographic Information Systems (GIS) provide a tool to explore spatial relationships within
and between data (Chang, 2016), and there is a growing trend of internet-based GIS (*i.e.,* GIS
designed for operating online over the World Wide Web) (Moretz, 2008). The application of internet
GIS through web mapping (the process of designing, generating, and delivering maps on the internet)
provides a number of advantages over traditional desktop-based GIS (Neumann, 2008). Web based
maps can deliver up to date data, can be generated using a low-cost software and hardware
infrastructure, and facilitate inexpensive map distributions. In addition, web mapping enables the
integration of different data sources and collaborative mapping (*e.g., Google Maps, Openstreet Maps*)
(Moretz, 2008; Neumann, 2008; Fu & Sun, 2010; Clarke, 2014). In the biodiversity conservation
discipline, web mapping offers greater accessibility and allows for user-driven interaction (Peterson,
2008). Furthermore, the advancement of smartphone applications (apps) that are linked to mobile web
based maps provides an avenue to involve broader audiences in the natural sciences and a convenient
tool for scientists to disseminate their research (Teacher *et al.*, 2013; Marchante *et al.*, 2017).
To take advantage of the potential of web-mapping, we developed a web-mapping application
for the Coral Triangle (CT) region of the Indo Pacific realm, a global hotspot for marine biodiversity
conservation due to its superlative species richness and endemicity (Hoeksema, 2007; Allen, 2008;
Veron *et al.*, 2009; Polidoro *et al.*, 2010; Walton *et al.*, 2014; Saeedi *et al.*, 2016). Because the region
has the highest density or marine species of anywhere in the ocean, it is a priority for marine
conservation. Furthermore, a large amount of biodiversity and natural resources data have been
collected for decades by scientists and numerous conservation programmes. However, data
repositories are scattered, and access to such data are limited. Previously, a systematic prioritization of
areas to include in an expanded Marine Protected Area (MPA) network was conducted, by
synthesizing data on biodiversity features data that are available for the region (Asaad. *et al*., 2018a;
2018b), but this alone does not make the information easily available to the public. To make this
important information more widely available to the general public and especially to policymakers, we
have designed three interlinked Web Geo-Apps that compile comprehensive and up-to-date
information on biodiversity features, areas of importance for biodiversity conservation, and priority
areas for expansion of the Coral Triangle MPA network. As an online atlas, these digital maps aim to
raise awareness of marine biodiversity conservation by making information about marine biodiversity,
marine protected areas, and areas of biodiversity importance both available to and accessible by the
public. This atlas can be used to support the application of marine biodiversity informatics in
conservation prioritization.



## 3. Methods (Web Map Design)

Digital maps were develop to interactively display geo-reference biodiversity information on the Coral Triangle (CT): (1) Biodiversity features; (2) Areas of importance for biodiversity conservation; and (3) Priority areas for Marine protected area (MPA) network expansion.

To generate the digital maps, related datasets were retrieved from the Coral Triangle database collected in the previous chapters (Asaad *et al*., 2018a; 2018b). These datasets were collated and and developed from various sources (Table 1). For consistency, all the datasets were clipped to the CT region following the implementation boundary of the Coral Triangle Initiative (CTI-CFF, 2009) with bounding geographic coordinates of $90^0$ E to $175^0$ E and $23^0$ N to $16^0$S. All of the data preparations were performed using ArcGIS Desktop 10.5 (ESRI, 2016a) and ArcGIS Pro. 2.0 (ESRI, 2017).

The ArcGIS Pro 2.0 were used to deliver and share all the maps to web feature layers in ArcGIS Online and designed these three digital maps using the ArcGIS Online template. Here, a similar template were used for each map to allow map comparisons. These digital maps used a website as an interface and can be accessed from any computers or other electronic devices that are connected to the internet using a standard browser (*e.g., Internet Explorer, Google Chrome or Safari*). The maps were hosted by the ArcGIS Online in a cloud service provided by the Amazon EC2 (Elastic Compute Cloud).

Each digital map consists of different feature layers:

- The map of "biodiversity features" is comprised of ten feature layers, including: (a) seven layers of biodiversity features (biogenic habitat, species richness-ranges, species richness-occurrence, species of conservation concern, species of restricted range, important areas for sea turtles and habitat rugosity; (b) two types of threat (anthropogenic and climate change); and (c) a composite of 16 environmental variables;
- The map of "areas of importance for biodiversity conservation" is comprised of two feature layers: (a) regional biodiversity hotspots; and (b) sites of biodiversity importance;
- The map of "priority areas for marine protected area network expansion" is consisted of nine feature layers: (a) three layers highlighting recommended priority areas for expansion of the Coral Triangle MPA network under scenarios of regional expansion to encompass 10%, 20% and 30% of CT marine area within the network; and (b) six layers showing priority areas for expansion of individual CT country MPA networks for Indonesia, Malaysia, the Philippines, Papua New Guinea, Solomon Islands and Timor Leste. Each layer of the national priority areas comprised of three scenarios of MPA expansion (10%, 20%, 30%);
- Three base layers are included for each web map: existing Marine Protected Areas, national Exclusive Economic Zones (EEZs) and country boundaries (Table 1).



To access the maps, a gallery-like web front page was developed with a hyperlink to each of the
digital map. Fifteen types of widgets (a control element in a graphical user interface) were embedded,
to allow users to explore a wide variety of functions offered by the maps (*e.g.,* Home button, Layer
list, Select, Draw *etc.*) (Table 2). A documentation website was developed to define the map's
objectives, datasets, classifications, and original citations of the sources.
**Table 1**. Coral Triangle datasets specifications.

| Data layer | Feature | Type, Spatial Resolution, Class | Descriptions | References |
|---|---|---|---|---|
| *1* | *2* | *3* | *4* | *5* |
| **Base Layers** | | | | |
| a. **Coral Triangle boundary** | Generated from the Coral Triangle Initiative Implementation boundary | Polygon | The boundary covers the full exclusive economic zones (EEZs) of Indonesia, Malaysia, Papua New Guinea, the Philippines, Solomon Islands, and Timor-Leste, and includes the EEZs of two additional nations: Brunei Darussalam and Singapore. | VLIZ, (2014) |
| b. **Country boundary** | Internal boundary of Coral Triangle countries | Polyline | The EEZ and internal boundaries are indicative only, and a dispute over boundaries exists. | VLIZ, (2014) |
| c. **Marine protected areas (MPA) coverage** | Coverage of 678 units of MPA | Polygon | The layers' attribute table provides detailed information following its native sources (WDPA, CTAtlas) (e.g., information of Name, Local Name, Designation Type, IUCN Category, coverage etc.) (IUCN & UNEP-WCMC,2016; Cros et al.,2014) with amendment and adjustment from local sources (Indonesian database). To allow simple indexing, a new CT MPAs ID format (MPA_ID) is introduced. The new ID consists of 10 digits: "A BC DEFG HIJ" <br><br>where:<br><br>▪ A = Country; 1 = Indonesia, 2 = Malaysia, 3 = Philippines, 4 = Papua New Guinea, 5 = Solomon Islands, and 6 = Timor Leste<br>▪ BC = IUCN MPAs Category; Strict Nature Reserve (1a = 11, 1b = 12), National Park (20), Habitat and Species Management Areas (40), Protected Landscape/Seascape (50) and Managed Resources Protected Areas (60)<br>▪ DEFG = Establishment year (*e.g.,* 1980)<br>▪ HIJ = Number; ordered based on their establishment year | IUCN & UNEP-WCMC (2016); Cros *et al.* (2014); MoF-MoMAF (2010); MoMAF (2016). |



|  | 1 | 2 | 3 | 4 | 5 |
|---|---|---|---|---|---|
| **Biodiversity Features** | | | | | |
| **a.** **Biogenic Habitat** | Spatial distribution of coral reef, seagrass and mangroves. | Grid square cells; 5 km; 3 classes | Calculated based on the number of biogenic habitat present in each cell. Cell values ranged from 1 – 3. | IMaRS-USF. & IRD., (2005); UNEP-WCMC et al., (2010); Giri et al., (2011a), (2011b); UNEP-WCMC & Short, (2005) |
| **b.** **Species richness - Ranges** | A modeled geographic distribution of 10,672 species ranges. | Grid square cells; 50 km; 10 classes | Calculated based on the number of predicted species in each cell. The number of predicted species per cell ranged from 0 to 5,509. | Kaschner *et al.* (2016) |
| **c.** **Species richness - Occurrence** | The occurrence records of 19,251 species. | Grid square cells; 50 km; 10 classes | Based on the index of expected species richness of $ES_{50}$ (estimated species in random 50 samples). | OBIS, (2015) |
| **d.** **Species of conservation concern** | The occurrence records of 834 species of conservation concern (Bony fish, anthozoans, elasmobranchs, mammals, and molluscs). | Grid square cells; 50 km; 10 classes | Based on the index of expected species richness of $ES_{35}$ (estimated species in random 50 samples). | OBIS, (2015); Froese & Pauly, (2016); IUCN (2015); UNEP-WCMC (2015) |
| **e.** **Species of restricted-range** | The distribution of 373 restricted-range reef fish species. | Grid square cells; 5 km; 10 classes | Calculated based on the number of species present in each cell. Cell values ranged from 1 – 101. | Allen, (2008); Allen & Erdmann, (2012) |
| **f.** **Important areas for sea turtle** | Nesting sites and migratory route of 6 species (2,055 records). | Grid square cells; 5 km; 3 classes | The richness calculated based on the number of sea turtle species present in each cell (*i.e.*, 1, 2, 3). | MoF-MoMAF, (2010); OBIS, (2015) |
| **g.** **Habitat rugosity** | A Vector Ruggedness Measure (VRM) of benthic terrain, generated from bathymetric data. | Grid square cells; 50 km; 10 classes | The VRM index ranged from 0.1 (areas with low terrain variations to 0.9 (areas with high terrain variations). | Basher *et al.*, (2014); Wright *et al.*, (2012) |
| **h.** **Anthropogenic Pressure (AP)** | Spatial distribution of AP on marine environments. | Grid square cells; 5 km; 10 classes | The cumulative impact of 19 different types of anthropogenic stressors. The AP value ranged from 0 – 15.4, indicating areas from low to high human-induced pressure. | Halpern *et al.*, (2008); Halpern *et al.*, (2015) |
| **i.** **Climate Change Pressure** | Spatial distribution of sea surface thermal stress level (the average of Degree Heating Weeks (DHW) from 2006 to 2099. | Grid square cells; 5 km; 10 classes | The projected thermal stress index ranged from 5.6 – 20.2, indicating areas from low to high vulnerability to climate change. | Van Hooidonk *et al.*, (2016) |




| | 1 | 2 | 3 | 4 | 5 |
|---|---|---|---|---|---|
| j. | **Environmental Variables** | Spatial distribution of environmental variables (physical, biochemical and nutrients). | Point; 50 km; 10 classes | Composite point features of 16 environmental variables, i.e., depth, slope, land distance, temperature, surface current, salinity, wind speed, tide, primary productivity, photosynthetically active radiation (PAR), chlorophyll-a, pH, dissolved oxygen, nitrate, silicate, and calcite. | Basher *et al.*, (2014). |
| **Areas of Importance for Biodiversity Conservation** | | | | | |
| a. | **Regional biodiversity hotspots** | Clusters of areas of biodiversity importance. | Grid square cells; 55 km; 3 classes of hotspots (high, medium and low) and 1 class not significant | Developed based on the multi-criteria analysis to five ecological criteria (sensitive habitat, species richness, the presence of species of conservation concern, the occurrence of restricted-range species, areas of importance for particular life history stages).<br><br>Analyzed based on the spatial patterns of data using the hotspots analysis tool in ArcGIS. The analysis clustered the cells from hotspot (high score cells) to coldspots (low score cells). | Asaad *et al.*, (2018a). |
| b. | **Sites of biodiversity importance** | Distribution of sites of areas of biodiversity importance. | Grid square cells; 55 km; 5 classes (high, medium-high, medium, medium-low and low) | Developed based on the similar ecological criteria to those used in the biodiversity hotspots region analysis.<br><br>While the hotspots analysis identified clustered areas of biodiversity importance. The site-based analysis identifies specific sites of highest biodiversity importance by analyzing the biodiversity score of each cell. The higher the score, the higher their biodiversity importance. | Asaad *et al.*, (2018a). |
| **Marine Protected Area (MPA) Network Expansion** | | | | | |
| a. | **Regional priority areas** | Spatial distribution of regional priority areas with three expansion scenario layers: 10%, 20% and 30%. | Grid square cells; 0.5 km | Prioritization analyses were performed using *Zonation* tools to analyze the proportions of the CT region placed into an MPA network (*e.g.,* expansion of the MPA network from existing coverage to 10%, 20% and 30 % of the Economic Exclusive Zone (EEZ) area).<br><br>The prioritization scenarios were based on seven sets of biodiversity features (biogenic habitat, habitat rugosity, species richness, distribution of threatened and endemic species, areas important for sea turtle); two types of threat (anthropogenic and climate change induced pressure); and the coverage of the existing MPA network.<br><br>Regional analyses were performed for the full CT EEZ region. | Asaad *et al.*, (2018a). |






| 1 | | 2 | 3 | 4 | 5 |
|---|---|---|---|---|---|
| a. | **National Priority Areas** | Spatial distribution of national priority areas with six layers of scenarios representing national MPA network expansion for Indonesia, Malaysia, the Philippines, Papua New Guinea, Solomon Islands and Timor Leste. | Grid square cells; 0.5 km | Developed based on the same approach as the regional priority areas.<br><br>National analyses were performed individually on each CT country national EEZ.<br><br>Each layer consisted of 3 scenarios of MPA expansion (10%, 20%, 30% ) | Asaad *et al.,* (2018b). |


**Table 2.** Widgets provided for the Coral Triangle Digital Maps.

| Icon | Widget | Functions |
|---|---|---|
| *1* | *2* | *3* |
| *Controller widgets  (Header panels)* | | |
|  | **About** | Displays general information about the apps, including purposes, data layers, and links to the documentation files. |
| | **Basemap Gallery** | Provides a gallery of base maps and allows users to select their preference. |
| | **Layer List** | Presents a list of layers in the map and allows users to interactively choose layers that need to be activated. Each layer has a checkbox and allows users to change the order of the layers in the map. |
| | **Legend** | Displays a legend of active layers showing in the map. |
| *Placeholder widgets (On-screen panel)* | | |
| | **Swipe** | Displays thumbnail views of a different layers at the on top of the map to enable a quick comparison of the content of different layers. Here we used the spyglass view model. |
| | **Draw** | Enables users to create and draw graphics (sketches) on the map. There are 11 feature creation tools (point, line, polyline, freehand, triangle, rectangle, circle, ellipse, polygon, freehand polygon, and text). It also displays measurement of the drawn features (lengths, areas, and perimeters). |
| | **Measurement** | Provides tools to measure areas (polygon), to calculate the distance (line), and to show the geographic coordinates (point). Each measurement can be displayed in a variety of measurement units (*i.e.,* metric and imperial system). |
| | **Print** | Provides service to print the map. This widget allows users to choose map layout and format (*e.g.,* pdf, jpg, gif) and an advanced option to select map scale, size and printing quality. |



| Table 2. continued | | |
|---|---|---|
| *1* | *2* | *3* |
|  | **Select** | Provides interactive tools to select features and perform tasks on the selected features. |
|  |  | There are four options to draw a selection: select by rectangle, polygon, circle, and line. |
|  |  | The selected features actions can be explored through: |
|  |  | ▪ Display tasks: Zoom to- , Pan to- and Flash. |
|  |  | ▪ Export: to CSV files, to feature collections, to GeoJSON (export to a features.geojson file). |
|  |  | ▪ Statistics: Display simple statistics of the selected features (sum, max, min, average, standard deviation). |
|  |  | ▪ Save to My Content: save selected features to My Content page in ArcGIS Online or ArcGIS Enterprises. |
|  |  | ▪ Create layer: enables to create layer for a single or selected feature. |
|  |  | ▪ View in Attribute Table: Previews the attribute table of the selected features. |
| *Off-panel Widgets* | | |
|  | **Home Button** | Displays the initial extent of the map. The bounding coordinates of the map is from 90$^0$E to 175$^0$E and 23$^0$N to 16$^0$S. |
|  | **Attribute table** | Shows a tabular view of operational layers' attributes. Located at the bottom of the map, and can be configured to display selected features, zoom to and filter the table based on the map extent. |
|  | **Coordinate** | Displays coordinates of the map (x and y values). Shows the coordinates in the WGS 1984 Mercator Auxiliary Sphere (WKID 3857) projection. Located at the lower-left corner of the map. |
|  | **Scale Bar** | Shows a scale bar of the map. Updated dynamically based on map's scale. Located in the lower-left corner of the map. |
|  | **My Location** | Displays the physical location and zooms the map to the users location. |
|  | **Zoom slider** | Provides an interactive zoom to the map display. |

## 4. Results

The digital map of the Coral Triangle is an online GIS database, and can be assessed through a web front-page (*www.marine.auckland.ac.nz/CTMAPS*) (Fig. 1). These geospatial datasets were built on three interlinked themes: (a) Biodiversity Features (*www.marine.auckland.ac.nz/CT_Biodiversity*) (Fig. 2), that provides comprehensive data on the region's marine protected areas, biodiversity features, threats and environmental characteristics; (b) Areas of Importance for Biodiversity Conservation (*www.marine.auckland.ac.nz/CT_Priority*) (Fig. 3), that provides spatial distributions of areas of high biodiversity conservation value; and (c) Priority areas for Coral Triangle Marine Protected Area (MPA) Network Expansion (*www.marine.auckland.ac.nz/CT_MPA*) (Fig. 4), that provides spatial information of priority areas for potential expansion of the existing MPA network. Relevant information on the maps can be accessed through an accompanying documentation website (*https://sites.google.com/view/coral-triangle-digital-map*) (Fig. 6).


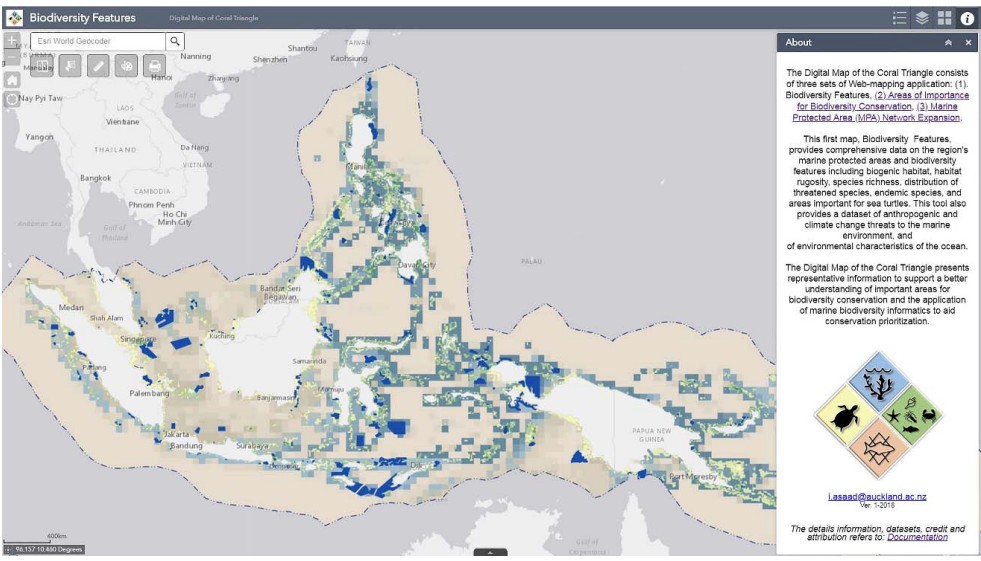

**Figure 1.** Coral Triangle web-mapping application front-page. This gallery-like interface provides a hyperlink to access each of the digital maps of the Coral Triangle.

**Figure 2.** The interface of the "Biodiversity Features" digital map. The right panel shows the "About" widgets that provides basic information about the map and hyperlinks to two other interrelated digital maps ("Areas of Biodiversity Importance" and "Priority Areas of MPA Network Expansion") and to the documentation file.






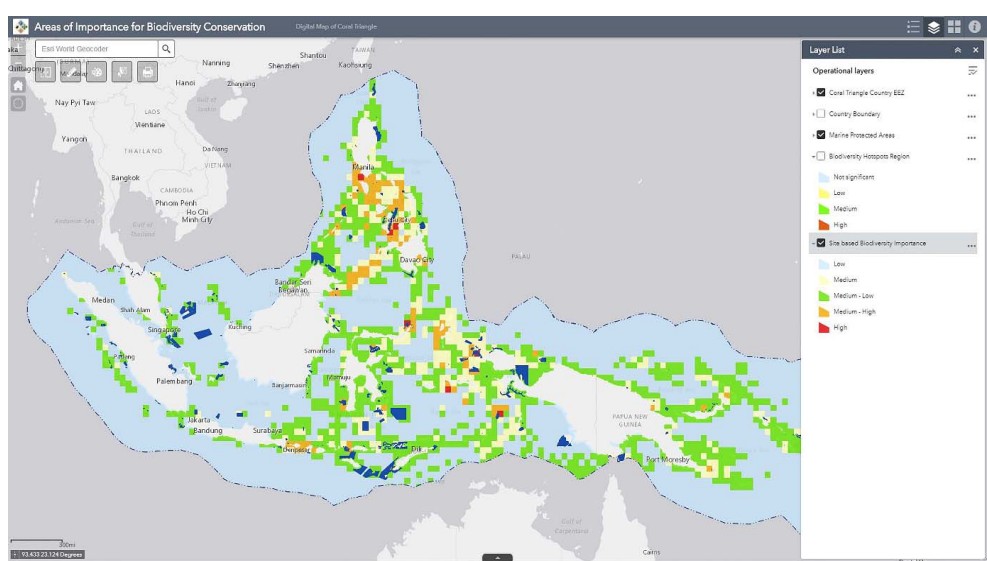

**Figure 3.** The interface of "Areas of Importance for Biodiversity Conservation" digital map. The right panel shows the "Layer List" widgets that provides access to interactively activated map layers and its accompanying map legends.

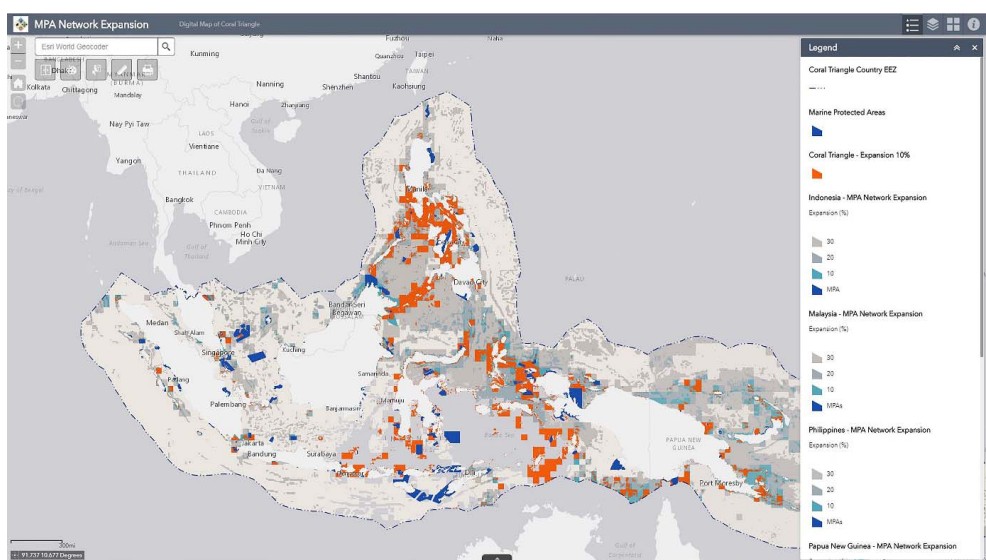

**Figure 4.** The interface of -"MPA Network Expansion" digital map showing priority areas for expansion of current MPAs or siting new ones, based upon analyses on chapter 4. The right panel shows the "Legend" widgets that display the map key including layer types (lines or polygon) and elements. The maps' screenshot shows 6 layers. Three layers each have one elements, and the other three are comprised of four different elements.



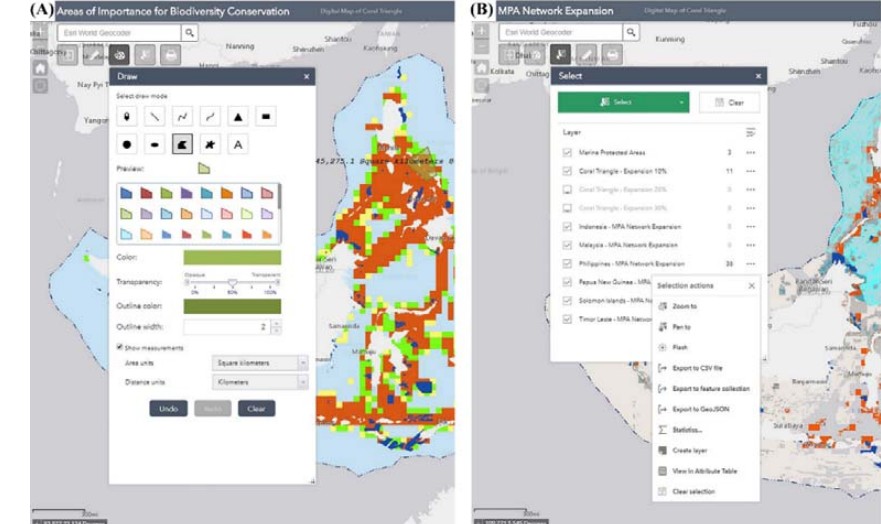

**Figure 5.** Interactive widgets: **(A)** "Draw" widgets, using various draw mode (e.g., point, line, polygon, freehand polygon), and colour scheme to sketch areas of interest in the map; **(B)** "Select" widgets, that allow user to select specific attributes and extract the selected spatial information in different formats.

## 5. Discussion

The digital maps of the Coral Triangle are designed to compile and showcase all of the currently-available marine biodiversity conservation data for the region and to give an overview of biodiversity conservation in the Coral Triangle region. They are derived from the most comprehensive biodiversity conservation datasets for the region, featuring spatial information for the region based on their habitat and species-specific attributes, vulnerabilities to threats, and environmental characteristics. The maps also include a set of data to indicate areas of importance for biodiversity conservation, existing MPAs, and priority areas of the designation of new MPAs or MPA expansion in the Coral Triangle, showing priorities for biodiversity conservation at both regional and national scales.

This study collated the datasets from open access sources with a variety of types and formats. Collating and comparing datasets from different sources presented a number of challenges. To have a consistent format and spatial attributes, all of the datasets were converted into a vector format (*i.e.,* lines or polygon shape), and standardized geographic projections. To reduce data discrepancy, the biodiversity feature datasets were classified using equal interval classes based on their biodiversity values. The datasets were then grouped into themes (biodiversity features, areas of important for biodiversity conservation and priority areas for MPA network expansion). Each theme consists of subthemes, to promote simple indexing, retrieving, and data management. Here, this study showed that to conduct biodiversity conservation programme, biodiversity data are indeed available, yet they are frequently scattered and not always easily accessed. Using an approach such as the one we describe here, these widely-scattered datasets can be integrated and amalgamated to perform a complex task such as biodiversity prioritization analysis (Asaad *et al.*, 2018a).





| 176 |
| 177 |
| 178 |
| 179 |
| 180 |
| 181 |
| 182 |
| 183 |
| 184 |
| 185 |
| 186 |

**Figure 6.** Documentation websites consisting of 4 panels (top right): Coral Triangle, Biodiversity Features, Areas of Importance for Biodiversity Conservation, MPA Network Expansion. Each panel is linked to specific information relating to the indicated map, including map objectives, datasets sources and specifications, data analysis and classifications. This file can be accessed from: *https://sites.google.com/view/coral-triangle-digital-map.*




This study developed an interactive web application that featured maps and geospatial contents using a
configurable template provided by ArcGIS Online. This approach reduces the complexity of code writing,
website programming and other technical knowledge needed to create a web map. We opted to use
accessible and less technical tools, to show that even with limited skills in internet GIS and web
development, scientific communities have an opportunity and develop a geospatial tool to support
biodiversity conservation. Replication of this type of approach in the other regions is important as there is a
continuing trend of biodiversity loss and limited resources are available to protect all of the important
biodiversity.
The digital maps were designed to enable an efficient decision-making process and to engage a wider
stakeholder audience. To support these objectives, all of the datasets were featured in a format that can be
overlaid and visualized directly using a standard web-browser. This web-browser platform facilitates
interactive access and examination of the data without the need for expensive GIS software. The spatial
information in each dataset can be extracted through: (a) intuitively hovering the mouse over and selecting a
feature; (b) using "select" widgets and exporting the selected features to preferred data formats; and (c)
reproducing the maps in suitable graphic formats using "print" widgets. The "select" widgets provides a
range of export formats, ranging from a generic "*comma-separated values (CSV)*" file that stores tabular data
in plain text, to a "*Geo JavaScript Object Notation (GeoJSON)*" file, an open standard format designed for
representing simple geographical features, along with their non-spatial attributes. The "print" widgets
provide an option to reproduce maps in a variety of formats such as pdf, jpeg, and gif, which facilitate
inclusion in presentations or embedding of maps in reports (Fig. 5). To enable and encourage data
explorations, the "select" widgets were supplied with functions to conduct simple statistical analysis (*e.g.,*
sum, average, maximum, minimum and standard deviation of selected data).
A previous online atlas of the CT was developed to support coral reef management and provided data
biophysical and MPA data from the region (Cros *et al.*, 2014). Though complementary in design, our digital
maps feature more systematic and comprehensive "biodiversity informatics" datasets. We collated integrated
ecological and biological datasets following a standard of ecological criteria to identify areas for biodiversity
conservation (Asaad, *et al.*, 2016). Our "Biodiversity Features" datasets are comprised of: biogenic habitat,
species richness-occurrences, species richness-ranges, species of conservation concern, restricted range
species, areas important for life history stage of species, and habitat rugosity. The datasets are ready to use
and are applicable for identifying areas priority areas for biodiversity conservation. In addition, this atlas
included datasets of threats, comprised of present anthropogenic and projected climate change induced
pressures. Knowledge of threat level provides key information for developing alternative marine spatial
planning and management strategies, *e.g.,* enforcement, habitat restoration, and mitigation (Green *et al.*,
2009; McLeod *et al.*, 2010; Maynard *et al.*, 2015). Furthermore, this digital maps also provided data for 16
environmental variables (including physical, chemical, and oceanographic variables). As such, this digital
map offers an opportunity to explore the relationship between biologically diverse areas and underlying
physical and chemical parameters, as well as the relationship with potential pressure factors.



The collections of geospatial data collated on this online GIS database are aimed to give access to
policymakers, scientific communities, and the general public full access to the most comprehensive, up-to-
date and integrated spatial information available for the Coral Triangle. This digital map presents
representative information to promote a better understanding of important areas for biodiversity conservation
and the application of marine biodiversity informatics to support conservation prioritization in the Coral
Triangle region.
**6.  Conclusion**
This atlas is a compendium of geospatial online and open-access data describing biodiversity
conservation in the Coral Triangle. It consists of 24 layers and three sets of interlinked digital maps: (a)
Biodiversity Features (providing comprehensive data on the region's marine protected areas, biodiversity
features, threats and environmental characteristics); (b) Areas of Importance for Biodiversity Conservation
(highlighting the spatial distribution of areas of high biodiversity conservation value); and (c) Marine
Protected Areas (MPA) Network Expansion (showing priority areas for expansion of current Coral Triangle
MPAs or siting of new MPAs). This publicly-accessible digital map provides the most comprehensive
biodiversity datasets available to date for the region and describes representative information to support a
better understanding of the key marine and coastal characteristics of the Coral Triangle.

**7. Author Contribution**
IA:  Conceiving the research ideas, designing the methodology, developing the web GIS applications and
writing of the manuscript.
CL: Advice and guidance in the study design, interpretation of the research, and reviewing the manuscript
for scientific rigor and readability.
ME: Advice and guidance in the study design, interpretation of the research, and reviewing the manuscript
for scientific rigor and readability.
MJ: Advice and guidance in the study design, interpretation of the research, and reviewing the manuscript
for scientific rigor and readability.
**8. Competing Interests**
"The authors declare that they have no conflict of interest."
**9. Acknowledgments**
IA is supported by New Zealand Aid Programme through New Zealand - ASEAN Scholarship. We
would like to thank Dr Tilmann Steinmetz (NIWA – Wellington), Keith Van Graafeiland (ESRI – Redlands),
Graeme Glen and Robert Carter (UoA – Auckland) for their technical assistance developing this digital map.

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
