# Peer review of "1.1. Coral Triangle – General Information"

_Earth System Science Data, 2018_

## Referee Comment (RC1) · Anonymous Referee #1 · 11 Aug 2018

Overall I find this paper to be of high quality in terms of its description of a comprehensive and useful online atlas for the Coral Triangle. In terms of well-established definitions of "online atlas" as applied to ocean and coast, aka "coastal web atlases," the authors might refer to the definitive work of Wright, Dwyer, and Cummins (2011) and reference it accordingly (especially in lines 51-53). Wright DJ, Dwyer E, & Cummins V eds (2011) Coastal Informatics: Web Atlas Design and Implementation (IGI-Global, Hershey, PA), p 350.

With that in mind the authors may want to rethink their title, as a digital map by itself is not necessarily equivalent to an online atlas. Technical a digital map could even be a pdf file that is placed online for someone to look at and download. But this falls very much short of what the authors intend. An online atlas is made up of a \*series\* of live,

**interactive** digital maps, as well as other resources (which could be static pdf files, downloadable data or data via web services, videos, etc.). Indeed, what the authors present is not just a single digital map, but an *interactive* online atlas allowing the user to make any number of digital maps (plural), as well as the useful web mapping **applications** for targeted uses, and involving more involved spatial analyses beyond the user choosing which layers to see on a map.

The data from which the user may create these maps is most excellent, as is the description of the various types and formats, as well as the technical details of the online atlas itself, especially as expressed in Tables 1 and 2. The figures are informative and highlight the very important contribution that this atlas will make to the marine conservation community by way of facilitating visualization and analysis or the most critical biodiversity features to consider (environmental variables, threats), the areas of biodiversity concern and conservation, and potential expansion of marine protected areas.

The paper could be improved by a brief statement or two much earlier in the paper as to the intended audience for the atlas (e.g., what particular organizations, governments, or initiatives). This might be most easily remedied by taking lines 209-223 and placing them in the introduction. These lines provide important history and context that should greet the reader earlier in the paper. And further, given the effort put into this atlas and its obvious power and utility, was it intended as a contribution to the Coral Triangle Initiative (e.g., Fidelman P, et al. (2014) Coalition cohesion for regional marine governance: A stakeholder analysis of the Coral Triangle Initiative. Ocean & Coastal Management 95(0):117-128). The paper does mention on line 72, the Coral Triangle MPA Network, so this might be equivalent. Perhaps make mention of these broader initiatives.

This atlas should most definitely be made known to the International Coastal Atlas Network, http://ican.iode.org, a program within the United Nations Intergovernmental Oceanographic Commission's International Oceanographic Data and Information Ex-

change (IODE). And to that end, the authors might make a comment or two about features of the atlas that show its good design for conservation and resource management audiences. The authors should consider reading and citing: Kopke K & Dwyer N (2017) ICAN - Best Practice Guide to Engage your Coastal Web Atlas User Community. IOC Manuals and Guides 75, IOC/2016/MG/75, Intergovernmental Oceanographic Commission of UNESCO, Paris, 35 pp., http://ican.iode.org/news/38-ican-cwa-user-interaction-guide

With regard to the Conclusion, this might be folded into the Discussion above and called "Discussion and Conclusion" unless in conflict with the journal's guidelines. As it stands, the "Conclusion" is more of a "Summary" rather than a conclusion. It merely restates the abstract and does not provide "conclusions" in terms of the outcome of an actual spatial analysis or even a benchmark/performance/use case review of your atlas, nor any recommendations as to its use. Also this final section, especially if folded into a combined "Discussion and Conclusion" section would be strengthened indeed by some recommendations. Digital atlases often suffer from a "build it and they will come" syndrome. This atlas is much too good for that, especially with the quality data and web apps available. Might the authors consider suggesting that it should it be part of the global Ocean Health Index effort, or perhaps supplements or boosts to any Conservation Intl, IUCN, or NIWA initiatives, membership in the ICAN and the broader efforts of the UN IOC IODE, or the GEO Marine Biodiversity Observation Network? How about a mention of the implications of this atlas for SDG 14 efforts and the upcoming UN Decade of Ocean Science? In other words, this reviewer is just suggesting some broader scope/broader utility statements for this paper beyond here is something wonderful that we built, to showcase yet another series of disparate datasets that were compiled.

With regard to minor technical suggestions: line 45 - references to works about GIS or internet/web GIS are in the hundreds now and the field changes rapidly. Given the emphasis of this paper on mapping and GIS for marine biodiversity conservation,
there are much more appropriate references than Chang (2016) for an intro to GIS and Moretz (2008). Instead of or in addition to Chang (2016) I would suggest: Wright DJ ed (2016) Ocean Solutions, Earth Solutions, 2nd Edition (Esri Press, Redlands, CA), 500 pp. and Hamylton SM (2017) Spatial Analysis of Coastal Environments (Cambridge University Press, Cambridge) p 290. And instead of Moretz (2008), Moretz (2017): Moretz, D, In Shekhar S, Xiong H, Zhou Z eds (2017), Encyclopedia of GIS, 2nd Edition (Springer Intl, Cham, Switzerland), 1074-1081. line 87 - ArcGIS Pro 2.0 is a singular entity, not plural line 258 - "... technical assistance [in] developing this digital map" should be "... technical assistance in developing this online atlas."

I look forward to seeing this paper in print, and best wishes to the authors for continued success.

---

## Referee Comment (RC2) · A. White (Referee) · 22 Aug 2018

A. White (Referee)

alan.white@sea-indonesia.org

This paper represents a very useful contribution to information base for the Coral Triangle countries. The work of Asaad et al. appears to be the development of an online "atlas" rather than just "digital maps" and it could thus be labeled as such. But, one consideration is how the current work as described by Asaad et al. interfaces with the Coral Triangle Atlas as developed by the Coral Triangle Support Partnership and is currently housed in the WorldFish Center in Penang Malaysia and under the direction of the Coral Triangle Initiative Secretariat based in Manado, Indonesia. In this regard, the current paper/work of Asaad probably needs to make mention of the CT Atlas and how a partnership might be considered or suggested as a recommendation.

Several questions about the datasets used that would be useful for the author to discuss in the paper are: 1. What are the gaps in the datasets used in terms of coverage of the coastal and marine areas in the CT? I would doubt there is continuous information for all of the coastline and thus a data layer that shows gaps would be very useful to see. This would help addresses bias introduced by present/absence of data for different geographical areas. 2. The authors might consider including the threat/climate data layers created by Reefs@Risk for Coral Triangle or at least explain why this data is not included? 3. Metadata: There is very little information about the data data used apart from the publications they come from which introduces biases and credibility issues that should at least be mentioned. 4. The mention of the Coral Triangle MPA Network (which doesn't really exist) should be clarified with reference to the Coral Triangle MPA System and Action Plan as described by the Coral Triangle Initiative publication of 2014. 5. People interested in this type of atlas/information are not average users but planners doing GIS at a broad scale who may want to access the raw data layers that the authors created. Thus making this available requires cleaning the data sets, filling in metadata and getting authorization which might be suggested for future work/recommendations? 6. Use of the word "biogenic" is not common and needs explaining. 7. Details of data such as species lists would be useful to expose. 8. Differences of scale of data need to be noted in some cases because scale makes some data layers not very compatible.

In the end, this work is very significant and will never be perfect and should be commended. Finally, I suggest the the conclusion be rewritten to include real "conclusions" and recommendations for the future work. As it stands, it is a brief summary of the work. Also, the sustainability of such an atlas is always difficult because they require resources to update and make them usable through time. In this regard the authors could make some suggestions about the main audience and how this will be updated and maintained. Also, partnerships with the CTI and CT Atlas could be suggested and pursued/recommended.

---

## Referee Comment (RC3) · S. Pittman (Referee) · 10 Sep 2018

This paper describes the data sets, post-processing of data, synthesis data and interactive tools for an online atlas of the Coral Triangle region. The manuscript is well written with just a few typographic and grammatical errors (e.g., tenses, plurals). The digital product will be useful to conservation and marine management by greatly improving access to a wide range of data that can support decision making in spatial planning and conservation prioritization. You mention it a little in the Discussion but it should be first set up in the Introduction section. Lastly, the conclusion section should be strengthened to include any limitations identified, major data gaps, next steps and ideas for how the data set and the tool can be applied and improved? Any additional functionality that would be useful?

[Figure]

My comments to help the authors revise the manuscript are below:

1. The Introduction section provides a good generic introduction to online databases and web maps but would benefit from a little more detail on the threats and conservation efforts in the Coral Triangle region that need better spatial data. Who exactly would benefit and why? Outline specific examples where this online map product will enhance decision making in the region? For example, in Line 189-192 of the Discussion you mention the importance of spatial prioritization and 'enable an efficient decision-making process'. It would be good if you introduce these projects and processes in the Introduction.

Table 1 was repeated again on pages 19-22 in my version of the PDF.

2. Line 60-61 - typo 3. Line 75 - typo with tense 'develop' should be 'developed' and 'geo-reference' should be 'geo-referenced' 4. Line 79 - typo 'and and' 5. Line 84 - ArcGIS Pro 2.0 was instead of 'The ArcGIS Pro 2.0 were 6. Line 85 - and design three instead of 'designed these three' 7. Line 86 - was used instead of 'were used' 8. Line 87 - computer or other electronic device connected to (no need for plurals) 9. Line 88 - hosted by ArcGIS Online not 'by the ArcGIS Online' 10. Line 95 - refer to Table 1 at the end of each bullet point where appropriate 11. Line 96-97 - Briefly mention how these layers were defined and provide citations 12. Line 98 - Briefly mention how these layers were defined and provide citations 13. Line 106 - The Table 1 has been referred to as 5.1 and occasionally a chapter has been referred to so please revise throughout to be consistent for this manuscript. Same for your figure numbers. It sounds like this manuscript was originally written as a chapter 14. Line 110 - see comment above 15. Line 110-111 - provide URL for documentation if available online 16. Table 2 - I don't think you need to show the widgets. These are generic to ESRI app developer and do not provide useful information. It is something that you expect to see in a user manual.

17. In your description of methods there is no mention of uncertainty or error in your datasets. The metadata should include information to allow the user to assess uncertainty. You should also provide discussion on this issue and on any perceived scale (temporal and spatial) limitations of the data in this manuscript. 18. Line 188 - to develop geospatial tools to support instead of 'and develop a geospatial tool' 19. Line 206 - typo - remove 'and' after 'provided' 20. Line 213 - typo - remove 'areas' before 'priority' 21. Line 217 - remove 'this' before 'digital maps' 22. 229-238 - The conclusion is basically just a repetition of the same information in the manuscript. It would be more useful for you to think about future applications (e.g., modelling biological distributions, predicting spatial change, mapping vulnerability to threats, spatial resilience) and improvements to tool functionality.

---

## Editor Comment (EC1) · F. Huettmann (Editor) · 14 Oct 2018

Dear Authors, I went through the reviews and found them positive, which reflects my own assessment of the work.

However, one issue I would like to stress is the notion of data, and "maps as data'. Arguably, a jpg or tiff image, or a map as a PDF figure, is less useful for people to use and re-use, and for transparent and repeatable science.

That's a certain problem in any Atlas work I know of, but can be addressed. Compliant ISO Metadata and Open Source tools, e.g. good database and GIS map formats, R and Open Source GIS tools can be key for that argument to convince and to show leadership.

[Figure]

Beyond the content and accuracy, I would like to emphasize the relevance of this topic for the wider community of users and for ESSD and its mission.

Thanks so much; I am looking forward to the response. Very best Falk Huettmann PhD, Professor Uni of Alaska Fairbanks

---

## Author Comment (AC1) · 5 Nov 2018

**1. Referee comments**

Overall I find this paper to be of high quality in terms of its description of a comprehensive and useful online atlas for the Coral Triangle. In terms of well-established definitions of "online atlas" as applied to ocean and coast, aka "coastal web atlases," the authors might refer to the definitive work of Wright, Dwyer, and Cummins (2011) and reference it accordingly (especially in lines 51-53). Wright DJ, Dwyer E, & Cummins V eds (2011) Coastal Informatics: Web Atlas Design and Implementation (IGI-Global, Hershey, PA), p 350.

**2. Response**

[Figure]

Thanks, we have added the definition of "online atlas" based on Wright at al (2011):

3. It is now read:

"For the coastal region, the application of web-atlases exists (e.g, Ireland Marine Atlas (atlas.marine.ie), Oregon Coastal Atlas (www.coastalatlas.net), The European Atlas (Barale et al 2015)). These coastal web-atlases have a variety of function such as serve as a web resources, data repository, interactive maps, and provide different type of geospatial analysis tools. Therefore, Wright et al (2011) define that the coastal web-atlas is "a collection of maps with supplementary tables, illustrations and information which systematically illustrate the coast"

1. Referee comments

With that in mind the authors may want to rethink their title, as a digital map by itself is not necessarily equivalent to an online atlas. Technical a digital map could even be a pdf file that is placed online for someone to look at and download. But this falls very much short of what the authors intend. An online atlas is made up of a *series* of live, **interactive** digital maps, as well as other resources (which could be static pdf files, downloadable data or data via web services, videos, etc.).

2. Response

We choose the term "Digital map" for our atlas as a "neutral title" that can be fit and explain all of the information within our atlas. Following reviewers suggestions and to make it consistent with the "series" and "interactive" function, we have replace the title to: An interactive atlas for marine biodiversity conservation in the Coral Triangle"

1. Referee comments

The paper could be improved by a brief statement or two much earlier in the paper as to the intended audience for the atlas (e.g., what particular organizations, govern-ments, or initiatives). This might be most easily remedied by taking lines 209-223 and placing them in the introduction. These lines provide important history and context that

should greet the reader earlier in the paper. And further, given the effort put into this atlas and its obvious power and utility, was it intended as a contribution to the Coral Triangle Initiative (e.g., Fidelman P, et al. (2014) Coalition cohesion for regional marine governance: A stakeholder analysis of the Coral Triangle Initiative. Ocean & Coastal Management 95(0):117-128). The paper does mention on line 72, the Coral Triangle MPA Network, so this might be equivalent. Perhaps make mention of these broader initiatives.

2. Response

We have revised the paragraph by moving the paragraph from Discussion sections and adding sentences of the objective of CTI that linked to our atlas

3. It is now read:

"To take advantage of the potential of web-mapping, here we developed a web-mapping application for the Coral Triangle (CT) region of the Indo Pacific realm, a global hotspot for marine biodiversity conservation due to its superlative species richness and endemicity (Hoeksema, 2007; Allen, 2008; Veron et al., 2009; Polidoro et al., 2010; Walton et al., 2014; Saeedi et al., 2016). Because the region has the highest density or marine species of anywhere in the ocean, it is a priority for marine conservation. Furthermore, a large amount of biodiversity and natural resources data have been collected for decades by scientists and numerous conservation programmes. However, data repositories are scattered, and access to such data are limited. Previously, a systematic geographic prioritization to develop Marine Protected Area (MPA) system was conducted (Asaad. et al., 2018a; 2018b), but this alone does not make the information easily available to the public. In addition, previous online atlas of the CT was developed to support coral reef management and provided biophysical and MPA data from the region (Cros et al., 2014), however an updated, more systematic and comprehensive "biodiversity informatics" datasets are required to showcase all of the available data in the region. Further, this web-atlas is aimed to support the objective of the

Interactive
comment

CTI-CFF initiative (the Coral Triangle Initiative on Coral Reefs, Fisheries and Food Security). The CTI-CFF initiative is a multilateral partnership of six countries (Indonesia, Malaysia, Papua New Guinea, the Philippines, Solomon Islands, and Timor-Leste) to working collaboratively to conserve and sustainably manage their coastal and marine resources (CTI-CFF, 2009, 2013). One of the objectives of this initiative is to establish and effectively manage Marine Protected Areas (MPA) network, which emphasizes the importance of developing and managing MPA throughout the region. In addition, an MPA system framework was developed to guide the development of network of MPAs in the region (Walton et al., 2014). Thus, the collections of geospatial data collated on this online GIS database are designed to support and assist in the development of marine protected areas and management of marine resources in the region. By giving an access to policymakers, scientific communities, and the general public to the most comprehensive, up-to-date and integrated spatial information available for the Coral Triangle"

**1. Referee comments**

This atlas should most definitely be made known to the International Coastal Atlas Network, ttp://ican.iode.org, a program within the United Nations Intergovernmental Oceanographic Commission's International Oceanographic Data and Information Exchange (IODE). And to that end, the authors might make a comment or two about features of the atlas that show its good design for conservation and resource management audiences. The authors should consider reading and citing: Kopke K & Dwyer N (2017) ICAN - Best Practice Guide to Engage your Coastal Web Atlas User Community. IOC Manuals and Guides 75, IOC/2016/MG/75, Intergovernmental Oceanographic Commission of UNESCO, Paris, 35 pp., http://ican.iode.org/news/38-ican-cwauser-interaction-guide. With regard to the Conclusion, this might be folded into the Discussion above and called "Discussion and Conclusion" unless in conflict with the journal's guidelines. As it stands, the "Conclusion" is more of a "Summary" rather than a conclusion. It merely restates the abstract and does not provide "conclusions" in terms of

the outcome of an actual spatial analysis or even a benchmark/performance/use case review of your atlas, nor any recommendations as to its use.

2. Response

We have revised the discussion and removed the conclusion section. With regards to broader initiatives (e.g., UN IOC IODE, the GEO Marine Biodiversity Observation Network, ICAN etc ), We have tried to contact and connect our Web Atlas to several initiatives. We will be very appreciated, if the reviewer introduce this atlas to your network as well.

3. We have added paragraph about future directions of the atlas:

"There are opportunities to improve and advance the geospatial functionality of this Coral Triangle atlas. An envisioned future version of this atlas is a dynamic online database which provides tools to add, upload and store new biodiversity data (e.g., species occurrence data). The growing trend of citizen science opens an opportunity to collect and integrate potentially massive amounts of data to fill gaps in the biodiversity data records. In addition, the availability of options to run online spatial analysis tasks such as identifying priority areas or delineating protected reserves in a defined geographic extent or for a specific dataset may offer an opportunity to further enhance the performance of this digital map.

In addition, the next step is to develop a network and connection to global initiative such as the IODE-ICAN (International Coastal Atlas Network), the Global Health Ocean Index (www.oceanhealthindex.org), the GEO-Marine Biodiversity Observation Network (boninabox.geobon.org), UNEP-WCMC Network (data.unep-wcmc.org) and others network related to the UN SDG 14 goal and the upcoming UN Decade of Ocean Science. This type of atlas potentially fills regional gaps data within such global initiatives and provide more details information that can be used to develop a region based biodiversity conservation strategy.

1. Referee comments

With regard to minor technical suggestions: line 45 - references to works about GIS or internet/web GIS are in the hundreds now and the field changes rapidly there are much more appropriate references than Chang (2016) for an intro to GIS and Moretz (2008). Instead of or in addition to Chang (2016) I would suggest: Wright DJ ed (2016) Ocean Solutions, Earth Solutions, 2nd Edition (Esri Press, Redlands, CA), 500 pp. and Hamylton SM (2017) Spatial Analysis of Coastal Environments (Cambridge University Press, Cambridge) p 290. And instead of Moretz (2008), Moretz (2017): Moretz, D, In Shekhar S, Xiong H, Zhou Z eds (2017), Encyclopedia of GIS, 2nd Edition (Springer Intl, Cham, Switzerland), 1074-1081.

2. Response

As suggested, we have amended the references from Chang (2016) to Wright et al (2016) and from Moretz (2008) to Moretz (2017).

1. Referee comments

line 87 - ArcGIS Pro 2.0 is a singular entity, not plural

line 258 - "... technical assistance [in] developing this digital map" should be "... technical assistance in developing this online atlas."

2. Response

We have corrected these grammatical errors

---

## Author Comment (AC2) · 5 Nov 2018

**1. Referee comments**

This paper represents a very useful contribution to information base for the Coral Triangle countries. The work of Asaad et al. appears to be the development of an online "atlas" rather than just "digital maps" and it could thus be labeled as such.

**2. Response**

We have revised the title to: "An Atlas of marine biodiversity conservation in the Coral Triangle"

1.Referee comments

[Figure]

But, one consideration is how the current work as described by Asaad et al. interfaces with the Coral Triangle Atlas as developed by the Coral Triangle Support Partnership and is currently housed in the WorldFish Center in Penang Malaysia and under the direction of the Coral Triangle Initiative Secretariat based in Manado, Indonesia. In this regard, the current paper/work of Asaad probably needs to make mention of the CT Atlas and how a partnership might be considered or suggested as a recommendation.

2. Response

We have added a paragraph in the Discussion section to describe the previous CT Atlas and how to link our atlas to the CT Atlas

3. It is now read:

"In the CT level, this atlas should be linked to the previous Coral Triangle Atlas developed by Cros et al. (2014) that are currently managed by the Coral Triangle Initiative Secretariat. These atlases are complementary in design and applications, and may provide an alternative options to the stakeholders to retrieve reliable Coral Triangle data. Here, our atlas provides a supplement and enriches the previous atlas by provide an access to explore area of biodiversity importance within the coral Triangle and a set of priority areas to designate new MPA within the region".

1.Referee comments

What are the gaps in the datasets used in terms of coverage of the coastal and marine areas in the CT? I would doubt there is continuous information for all of the coastline and thus a data layer that shows gaps would be very useful to see. This would help addresses bias introduced by present/absence of data for different geographical areas.

2. Response

One of the data gaps of this atlas is related to habitat distribution maps. We have added paragraph in the discussion section to explain about the limitation of this atlas, including gaps in the habitat distribution maps.

3. Amended paragraph

In this atlas, the biogenic habitat distribution map was retrieved from three types of coastal habitats (coral reefs, seagrass, and mangroves), that may generate a biased toward coastal region. In the absence of other habitats data (e.g. soft sediments, rocky inter-tidal zones and other sub-tidal habitats), this atlas introduces layer of benthic terrain rugosity as a proxy of habitat heterogeneity. This benthic rugosity layers covers the entire CT region (beyond the coastal areas, for which distribution data were available). However, a detailed habitat maps and a defined list of habitat types are needed to develop a comprehensive biodiversity conservation programme"

1.Referee comments

The authors might consider including the threat/climate data layers created by Reefs@Risk for Coral Triangle or at least explain why this data is not included?

2. Response

We used recent datasets of climate data from Hooidonk et al. 2016. The Hooidonk dataset provide more details dataset and cover entire regions (not only in coral area).

Van Hooidonk, R., Maynard, J., Tamelander, J., Gove, J., Ahmadia, G., Raymundo, L., Williams, G., Heron, S.F., & Planes, S. (2016). Local-scale projections of coral reef futures and implications of the Paris Agreement. Scientific reports, 6, 39666

1.Referee comments

Metadata: There is very little information about the data data used apart from the publications they come from which introduces biases and credibility issues that should at least be mentioned.

2. Response

We have added the matadata in the supplementary sections. For your considerations, we have attached the metadata file.
1.Referee comments

The mention of the Coral Triangle MPA Network (which doesn't really exist) should be clarified with reference to the Coral Triangle MPA System and Action Plan as described by the Coral Triangle Initiative publication of 2014.

2. Response

We have amended the paragraph and replace all of the MPA network to MPA system, and we added a paragraph to explain the the current MPA system within the CT.

3. It is now read:

"One of the objectives of the Coral Triangle Initiative is to establish and effectively manage MPA within the region. This objective has supported by a target: to develop a fully functional region-wide Coral Triangle MPA System (CTMPAS) (CTI-CFF, 2009). The CTMPAS is a system of MPA within the CT which include a range of MPA types and MPA network. This system comprises of individual MPA or MPA network that form local ecological and/or governance networks that are nested within larger-scale of social networks. The MPA systems has introduced due to the expanse of the Coral Triangle, thus it is not considered feasible to develop a regional ecologically connected network that cover all of the Coral Triangle region (CTI-CFF, 2013)"

1.Referee comments

People interested in this type of atlas/information are not average users but planners doing GIS at a broad scale who may want to access the raw data layers that the authors created. Thus making this available requires cleaning the data sets, filling in metadata and getting authorization which might be suggested for future work/recommendations?

2. Response

We agree that one of the potential users of this atlas is advanced GIS user. We have provided a metadata to all of the dataset, including information on the data sources of

raw data.

1.Referee comments

Use of the word "biogenic" is not common and needs explaining.

2. Response

We have explained the term "biogenic" in the Methods. Biogenic habitat refers to the habitats that are created by plants and animals.

1.Referee comments

Details of data such as species lists would be useful to expose.

2. Response

As all of the data are part of our previous paper that are published in the different journal, to fully access and to explore the raw species data, we have provided a full citation of original data in the Table 1. We also provide a complete information in the metadata.

1.Referee comments

Differences of scale of data need to be noted in some cases because scale makes some data layers not very compatible.

2. Response

We have noted the spatial resolutions of each dataset in the table 1.

1.Referee comments

Finally, I suggest the the conclusion be rewritten to include real "conclusions" and recommendations for the future work. As it stands, it is a brief summary of the work. Also, the sustainability of such an atlas is always difficult because they require resources to update and make them usable through time. In this regard the authors could make

some suggestions about the main audience and how this will be updated and maintained. Also, partnerships with the CTI and CT Atlas could be suggested and pursued / recommended.

2. Response

We have revised and removed the conclusion section and folded it into Discussion and Future Directions. The last section of Discussion has explained about the future directions.

3. Amended paragraph:

"There are opportunities to improve and advance the geospatial functionality of this Coral Triangle atlas. An envisioned future version of this atlas is a dynamic online database which provides tools to add, upload and store new biodiversity data (e.g., species occurrence data). The growing trend of citizen science opens an opportunity to collect and integrate potentially massive amounts of data to fill gaps in the biodiversity data records. In addition, the availability of options to run online spatial analysis tasks such as identifying priority areas or delineating protected reserves in a defined geographic extent or for a specific dataset may offer an opportunity to further enhance the performance of this digital map.

In addition, the next step is to develop a network and connection to global initiative such as IODE-ICAN (International Coastal Atlas Network), the Global Health Ocean Index (www.oceanhealthindex.org), the GEO-Marine Biodiversity Observation Network (boninabox.geobon.org), UNEP-WCMC Network (data.unep-wcmc.org) and others network related to the UN SDG 14 goal and the upcoming UN Decade of Ocean Science. This type of atlas potentially fills regional gaps data within such global initiatives and provide more details information that can be used to develop a region based biodiversity conservation strategy"

Please also note the supplement to this comment:

https://www.earth-syst-sci-data-discuss.net/essd-2018-80/essd-2018-80-AC2-supplement.pdf

[Figure]

**Supplement:**

**Supplementary Materials**
**Asaad et al.,**

**2. Metadata of the Atlas of biodiversity conservation in the Coral Triangle**

| Biogenic Habitat | |
| --- | --- |
| Description | This map presents a spatial distribution of three biogenic habitat (coral reef, seagrass and mangrove) and shows the habitat richness at each cell. |
| Temporal range | Refers to the native sources |
| Geographical range | Coral Triangle of the Indo Pacific Realm |
| Citations | Asaad, I., Lundquist, C. J., Erdmann, M. V., & Costello, M. J. (2018). Delineating priority areas for marine biodiversity conservation in the Coral Triangle. *Biological Conservation, 222*, 198-211. doi: dx.doi.org/10.1016/j.biocon.2018.03.037 |
| Original datasets / citations | ▪ Coral Reef Distribution: IMaRS-USF. and IRD. (2005); UNEP-WCMC et al. (2010) (http://data.unep-wcmc.org/datasets/13)
 ▪ Seagrass Distribution: UNEP-WCMC and Short (2005) (http://data.unepwcmc.org/datasets/10 and and data.unep-wcmc.org/datasets/9
 ▪ Mangrove Distribution: Giri *et al*. (2011a, 2011b) (http://data.unep-wcmc.org/datasets/21) |
| Purpose of creation | To identify areas of biodiversity importance within the Coral Triangle based on habitat richness. |
| Creation methodology | All datasets were clipped to the Coral Triangle region using a grid approach of 5 km cells. Using this approach, the datasets were presented and mapped into a regular shape of a grid square. Thus, those three datasets were superimposed and overlaid to generate a single dataset. Further, the dataset was classified and scored based on the total number of habitats that fell within each cell. Cell values ranged from 1 – 3. The methodology is fully described at Asaad et al (2018). |
| Version | 1 (July 2018) |
| Keywords: | Coral Triangle, biodiversity importance, biodiversity feature, biogenic habitat. |
| Category | Biodiversity Features |
| Limitations: | The biogenic habitat distribution map was retrieved from three types of coastal habitats (coral reefs, seagrass, and mangroves), that may generate a biased toward coastal region.
 A detailed habitat maps and a defined list of habitat types are needed to develop a comprehensive biodiversity conservation programme. |
| Main access/use constraint: | Creative Commons Attribution 4.0 (CC BY 4.0) |
| Contact Organization | Institute of Marine Science, University of Auckland. |
| Name | Irawan Asaad |
| City | Auckland |
| Country | New Zealand |
| email | i.asaad@auckland.ac.nz |
| Data format | Geodatabase (Grid square cells; polygon) |
| Distribution format | GeoJSON |
| Dataset size | 0.65 MB |
| Webpage | *www.marine.auckland.ac.nz/CT_Biodiversity* |

| | |
|---|---|
| Otherweb page | https://uoa.maps.arcgis.com/apps/webappviewer/index.html?id=1406b9131245493195c12a1df3d2ada6 |
| Resolution / scale | 5 km |
| Reference System | WGS 84 |
| North bounding | 22.0 |
| South bounding | -16.0 |
| West bounding | 90.0 |
| East bounding | 175.0 |
| Date of metadata | 10th July 2018 |

| Species Richness Ranges | |
|---|---|
| Description | This map shows the potential species richness based on the modelled geographic species ranges extracted from 10,672 species ranges that were retrieved from AquaMaps database (www.aquampas.org). The richness was calculated based on the total number of species ranges that fell within the cell. |
| Temporal range | Follow the native sources |
| Geographical range | Coral Triangle of the Indo Pacific Realm |
| Citations | Asaad, I., Lundquist, C. J., Erdmann, M. V., & Costello, M. J. (2018). Delineating priority areas for marine biodiversity conservation in the Coral Triangle. *Biological Conservation, 222*, 198-211. doi: dx.doi.org/10.1016/j.biocon.2018.03.037 |
| Original datasets / citations | Kaschner, K., Rius-Barile, J., Kesner-Reyes, K., Garilao, C., Kullander, S.O., Rees, T., Froese, R. (2016). AquaMaps: Predicted Range Maps for Aquatic Species. Worldwide Web Electronic Publication. www.aquamaps.org (Version 08/2016). |
| Purpose of creation | To identify areas of biodiversity importance within the Coral Triangle based on species richness. |
| Creation methodology | To assess the criterion of species richness, a modelled geographic species ranges extracted from AquaMaps (Kaschner et al., 2016). AquaMaps generates a prediction of relative probabilities of species range at a resolution of half-degree cells. Each cell contains a probability value ranging from 0 and 1, representing the relative suitability of that cell for the specified species. The richness was based on the number of predicted species in each cell. Within the study area, the number of species per 0.5° cells ranged from 11 to 5509. Thus, the cells were classified into 10 equal interval classes based on the total number of species that fell within each cell, i.e., class 1 (11–550 species); class 2 (>550 – 1.100) to class 10 (>4950 – 5509 species).
The methodology is fully described at Asaad et al (2018) |
| Version | 1 (July 2018) |
| Keywords: | Coral Triangle, biodiversity importance, biodiversity feature, species richness. |
| Category | Biodiversity Features |
| Limitations: | The spatial resolution of the map is 50 km. The predicted extent is based on species occurrence record and environmental distribution modelling. As a modelling approach, the present distribution needs a confirmation from field observations. In addition, the data mostly available for a common species. |

| | |
|---|---|
| Main access/use constraint: | Creative Commons Attribution 4.0 (CC BY 4.0) |
| Contact Organization | Institute of Marine Science, University of Auckland |
| Name | Irawan Asaad |
| City | Auckland |
| Country | New Zealand |
| email | i.asaad@auckland.ac.nz |
| Data format | Geodatabase (Grid square cells; polygon) |
| Distribution format | GeoJSON |
| Dataset size | 1.98 MB |
| Webpage | *www.marine.auckland.ac.nz/CT_Biodiversity* |
| Otherweb page | https://uoa.maps.arcgis.com/apps/webappviewer/index.html?id=1406b9131245493195c12a1df3d2ada6 |
| Resolution / scale | 50 km |
| Reference System | WGS 84 |
| North bounding | 22.0 |
| South bounding | -16.0 |
| West bounding | 90.0 |
| East bounding | 175.0 |
| Date of metadata | 10th July 2018 |

| Species Richness- Occurrence | |
|---|---|
| Description | This map presents the potential species richness based on the occurrence records of 19,251 species retrieved from OBIS datasets (www.iobis.org). The richness was analysed based on the Hulbert's index of expected species richness of ES50 (estimated species in random 50 samples). |
| Temporal range | Follow the native sources |
| Geographical range | Coral Triangle of the Indo Pacific Realm |
| Citations | Asaad, I., Lundquist, C. J., Erdmann, M. V., & Costello, M. J. (2018). Delineating priority areas for marine biodiversity conservation in the Coral Triangle. *Biological Conservation, 222*, 198-211. doi: dx.doi.org/10.1016/j.biocon.2018.03.037 |
| Original datasets / citations | OBIS (2015). Data from the Ocean Biogeographic Information System. Intergovernmental Oceanographic Commission of UNESCO. Retrieved 02/05/2015. http://www.iobis.org |
| Purpose of creation | To identify areas of biodiversity importance within the Coral Triangle based on species richness. |
| Creation methodology | To assess the of species richness, the occurrence records of 19,251 species were retrieved from OBIS datasets (www.iobis.org). The richness was analysed based on the Hulbert's index of expected species richness of ES50 (estimated species in random 50 samples). Using 0.5 degree cells, the richness was based on the index of estimated species in each cell. Within the study area, the species index ranged from 0 – 50. Thus, the cells were classified into 10 equal interval classes, i.e., class 1 (ES50 1–5); class 2;(5−10) to class 10 (45–50). The methodology is fully described at Asaad et al (2018) |

| | |
|---|---|
| Version | 1 (July 2018) |
| Keywords: | Coral Triangle, biodiversity importance, biodiversity feature, species richness. |
| Category | Biodiversity Features |
| Limitations: | The occurrence records are mostly available for a wide-ranging species, and are likely prone to omission errors (false negatives). |
| Main access/use constraint: | Creative Commons Attribution 4.0 (CC BY 4.0) |
| Contact Organization | Institute of Marine Science, University of Auckland |
| Name | Irawan Asaad |
| City | Auckland |
| Country | New Zealand |
| email | i.asaad@auckland.ac.nz |
| Data format | Geodatabase (Grid square cells; polygon) |
| Distribution format | GeoJSON |
| Dataset size | 0.06 MB |
| Webpage | www.marine.auckland.ac.nz/CT_Biodiversity |
| Otherweb page | https://uoa.maps.arcgis.com/apps/webappviewer/index.html?id=1406b9131245493195c12a1df3d2ada6 |
| Resolution / scale | 50 km |
| Reference System | WGS 1984 |
| North bounding | 22.0 |
| South bounding | -16.0 |
| West bounding | 90.0 |
| East bounding | 175.0 |
| Date of metadata | 10th July 2018 |

| Species of Conservation Concern | |
|---|---|
| Description | This map shows the distribution of species of conservation concern based on the occurrence records of 834 species (Bony fish, anthozoans, elasmobranchs, mammals, and molluscs) retrieved from OBIS datasets (www.iobis.org). The richness was analysed based on the Hulbert's index of expected species richness of ES35 (estimated species in random 35 samples). |
| Temporal range | Refers to the native sources. |
| Geographical range | Coral Triangle of the Indo Pacific Realm |
| Citations | Asaad, I., Lundquist, C. J., Erdmann, M. V., & Costello, M. J. (2018). Delineating priority areas for marine biodiversity conservation in the Coral Triangle. *Biological Conservation, 222*, 198-211. doi: dx.doi.org/10.1016/j.biocon.2018.03.037 |
| Original datasets / citations | OBIS (2015). Data from the Ocean Biogeographic Information System. Intergovernmental Oceanographic Commission of UNESCO. Retrieved 02/05/2015. http://www.iobis.org |

| | |
|---|---|
| | Froese, R., Pauly, D. (2016). FishBase. World Wide Web Electronic Publication. Retrieved version (06/2016). www.fishbase.org. |
| Purpose of creation | To identify areas of biodiversity importance within the Coral Triangle based on the distribution of species of conservation concern. |
| Creation methodology | The distribution of species of conservation concern were evaluated based on species occurrence records of five classes on each 0.5. degree cell. The occurrence records were extracted from OBIS (www.iobis.org) and FishBase (www.fishbase.org). A species was included as species of conservation concern as recognized by IUCN Red List categories (IUCN, 2015), CITES (UNEP-WCMC, 2015) and national directives of the Coral Triangle countries (Indonesia, Malaysia and The Philippines). A Hulbert index with ES35 (estimated species in random 35 samples) was used to identify cells with the highest richness of species of conservation concern. The cells were classified into 10 equal interval classes, i.e., class 1 (ES35 1–4); class 2 (>4 – 7) to class 10 (.> 28–35). The methodology is fully described at Asaad *et al* (2018). |
| Version | 1 (July 2018) |
| Keywords: | Coral Triangle, biodiversity importance, biodiversity feature, biogenic habitat, species of conservation concern |
| Category | Biodiversity Features |
| Limitations: | The dataset covers only selected taxa to represent a diversity of threatened taxa. Thus, not all of the threatened species that may exist within the region were listed in the maps. |
| Main access/use constraint: | Creative Commons Attribution 4.0 (CC BY 4.0) |
| Contact Organization | Institute of Marine Science, University of Auckland |
| Name | Irawan Asaad |
| City | Auckland |
| Country | New Zealand |
| email | i.asaad@auckland.ac.nz |
| Data format | Geodatabase (Grid square cells; polygon) |
| Distribution format | GeoJSON |
| Dataset size | 0.18 MB |
| Webpage | *www.marine.auckland.ac.nz/CT_Biodiversity* |
| Otherweb page | https://uoa.maps.arcgis.com/apps/webappviewer/index.html?id=1406b9131245493195c12a1df3d2ada6 |
| Resolution / scale | 50 km |
| Reference System | WGS 84 |
| North bounding | 22.0 |
| South bounding | -16.0 |
| West bounding | 90.0 |
| East bounding | 175.0 |
| Date of metadata | 10th July 2018 |

| Species of Restricted-range (Endemic species) | |
|---|---|
| Description | This map shows the distribution of restricted range species, based on the ranges of 373 reef fishes that are known to be endemic to the Coral Triangle. The data was extracted from a dataset of nearly 4000 species of Indo-Pacific reef fishes (Allen, 2008; Allen and Erdmann, 2013). The richness was calculated based on the total number of species ranges that fell within the polygon. |
| Temporal range | Refers to the native sources |
| Geographical range | Coral Triangle of the Indo Pacific Realm |

| | |
|---|---|
| Citations | Asaad, I., Lundquist, C. J., Erdmann, M. V., & Costello, M. J. (2018). Delineating priority areas for marine biodiversity conservation in the Coral Triangle. *Biological Conservation, 222*, 198-211. doi: dx.doi.org/10.1016/j.biocon.2018.03.037 |
| Original datasets / citations | Allen, G.R., Erdmann, M.V., 2013. Reef Fishes of the East Indies. Mobile Application Software. Version 1.1 (Rev.10.2016). Retrieved 15/06/2016. https://geo.itunes.apple.com/us/app/reef-fishes-east-indies-vol./id705188551?mt=8. |
| Purpose of creation | To identify areas of biodiversity importance within the Coral Triangle based on the distribution of restricted range reef fishes species. |
| Creation methodology | The distribution of restricted-range species was assessed using the distributions of 373 reef fishes (comprising 150 genera and 47 families) that are each endemic to the Coral Triangle region. The ranges of reef fishes species were assigned to 5 km grid cells. The richness was calculated based on the total number of restricted range reef fishes that fell within each cell. The value ranged from 0 to 101 species. Thus, the cells were classified based on an equal interval into 10 classes i.e.: class 1 (1–10 species); class 2 (10–20 species) to class 10 (100 – 101 species).
 The methodology is fully described at Asaad *et al* (2018) |
| Version | 1 (July 2018) |
| Keywords: | Coral Triangle, biodiversity importance, biodiversity feature, species restricted range, endemic spcies |
| Category | Biodiversity Features |
| Limitations: | This dataset uses predefined descriptions:
 ▪ Restricted-range is defined as a reef fish species with a spatial distribution of<5 million $km^2$ and whose known range is only within the Coral Triangle.
 ▪ Reef fishes is defined as fish species that live on shallow water coral reefs and associated substrata (i.e., sand or rubble patches, seagrass beds, etc.) <60m deep.
 Other datasets and references may apply different definitions and descriptions. |
| Main access/use constraint: | Creative Commons Attribution 4.0 (CC BY 4.0) |
| Contact Organization | Institute of Marine Science, University of Auckland |
| Name | Irawan Asaad |
| City | Auckland |
| Country | New Zealand |
| email | i.asaad@auckland.ac.nz |
| Data format | Geodatabase (Grid cells; polygon) |
| Distribution format | GeoJSON |
| Dataset size | 0.54 MB |
| Webpage | *www.marine.auckland.ac.nz/CT_Biodiversity* |
| Otherweb page | https://uoa.maps.arcgis.com/apps/webappviewer/index.html?id=1406b9131245493195c12a1df3d2ada6 |
| Resolution / scale | 5 km |
| Reference System | WGS 84 |
| North bounding | 22.0 |
| South bounding | -16.0 |
| West bounding | 90.0 |
| East bounding | 175.0 |
| Date of metadata | 10th July 2018 |

| | Areas important for Sea Turtle |
|---|---|
| Description | This map presents the distribution of nesting sites and migratory routes of six species of sea turtle, and shows the richness at each cell. The richness was calculated based on the total number of species that fell within the cell. |
| Temporal range | Refers to the native sources |
| Geographical range | Coral Triangle of the Indo Pacific Realm |
| Citations | Asaad, I., Lundquist, C. J., Erdmann, M. V., & Costello, M. J. (2018). Delineating priority areas for marine biodiversity conservation in the Coral Triangle. *Biological Conservation, 222*, 198-211. doi: dx.doi.org/10.1016/j.biocon.2018.03.037 |
| Original datasets / citations | OBIS (2015). Data from the Ocean Biogeographic Information System. Intergovernmental Oceanographic Commission of UNESCO. Retrieved 02/05/2015. http://www.iobis.org

MoF-MoMAF, 2010. Ecological Representation Gap Analysis for Conservation Areas in Indonesia. Ministry of Forestry and Ministry of Marine Affairs and Fisheries, Jakarta- Indonesia. |
| Purpose of creation | To identify areas of biodiversity importance within the Coral Triangle based on the criterion of areas importance for a life history stages of species. |
| Creation methodology | Sea turtle nesting habitat and migratory routes were used as indicators of important areas for sea turtles. Six sea turtle species inhabit the Coral Triangle: green, leatherback, loggerhead, hawksbill, olive Ridley and flatback turtles. A total of 2055 point occurrence records of sea turtles were retrieved from OBIS (www.iobis.org) and Indonesian sea turtle datasets (MoF-MoMAF, 2010). The occurrence points were transformed into grid cells of 5 km. Thus, the richness was calculated based on the total number of species that fell within each cell. The value of each cell ranged from 0 to 3 species. No cells had more than three species of turtle present.
The methodology is fully described at Asaad *et al* (2018) |
| Version | 1 (July 2018) |
| Keywords: | Coral Triangle, biodiversity importance, biodiversity feature, species restricted range, endemic species |
| Category | Biodiversity Features |
| Limitations: | This dataset uses a point location for sea turtle nesting area and migratory route. Without an exact boundary of nesting beaches or the migratory perimeter thus this dataset prone to an omission errors. |
| Main access/use constraint: | Creative Commons Attribution 4.0 (CC BY 4.0) |
| Contact Organization | Institute of Marine Science, University of Auckland |
| Name | Irawan Asaad |
| City | Auckland |
| Country | New Zealand |
| email | i.asaad@auckland.ac.nz |
| Data format | Geodatabase (Grid square cells; polygon) |
| Distribution format | GeoJSON |
| Dataset size | 0.17 MB |
| Webpage | *www.marine.auckland.ac.nz/CT_Biodiversity* |
| Otherweb page | https://uoa.maps.arcgis.com/apps/webappviewer/index.html?id=1406b9131245493195c12a1df3d2ada6 |
| Resolution / scale | 5 km |

| | |
|---|---|
| Reference System | WGS 84 |
| North bounding | 22.0 |
| South bounding | -16.0 |
| West bounding | 90.0 |
| East bounding | 175.0 |
| Date of metadata | 10th July 2018 |

| **Environmental Variables** | |
|---|---|
| Description | This map shows spatial distribution of environmental variables (physical, biochemical and nutrients). This is a composite of point futures of 16 environmental variables, i.e., depth, slope, land distance, temperature, surface current, salinity, wind speed, tide, primary productivity, photosynthetically active radiation (PAR), chlorophyll-a, pH, dissolved oxygen, nitrate, silicate, and calcite. |
| Temporal range | Refers to the native sources |
| Geographical range | Coral Triangle of the Indo Pacific Realm |
| Citations | Asaad, I., Lundquist, C. J., Erdmann, M. V., & Costello, M. J. (2018). Delineating priority areas for marine biodiversity conservation in the Coral Triangle. *Biological Conservation, 222*, 198-211. doi: dx.doi.org/10.1016/j.biocon.2018.03.037 |
| Original datasets / citations | Basher, Z., Bowden, D.A., Costello, M.J., 2014. Global Marine Environment Datasets (GMED)-World Wide Web electronic publication. Version 1.0 (Rev.01.2014). Retrieved 15/01/2016. http://gmed.auckland.ac.nz. |
| Purpose of creation | To describe environmental characteristics of the Coral Triangle |
| Creation methodology | 16 Environmental variables were extracted from the Global Marine Environment Datasets (GMED). Thus, the data were transformed into a point dataset. A composite point dataset was generated to with a spatial resolution of 50 km. The methodology is fully described at Asaad *et al* (2018) |
| Version | 1 (July 2018) |
| Keywords: | Coral Triangle, biodiversity importance, biodiversity feature, environmental variables |
| Category | Biodiversity Features |
| Limitations: | |
| Main access/use constraint: | Creative Commons Attribution 4.0 (CC BY 4.0) |
| Contact Organization | Institute of Marine Science, University of Auckland |
| Name | Irawan Asaad |
| City | Auckland |
| Country | New Zealand |
| email | i.asaad@auckland.ac.nz |
| Data format | Geodatabase (point) |

| | |
|---|---|
| Distribution format | GeoJSON |
| Dataset size | 3.59 MB |
| Webpage | *www.marine.auckland.ac.nz/CT_Biodiversity* |
| Otherweb page | https://uoa.maps.arcgis.com/apps/webappviewer/index.html?id=1406b9131245493195c12a1df3d2ada6 |
| Resolution / scale | 50 km |
| Reference System | WGS 84 |
| North bounding | 22.0 |
| South bounding | -16.0 |
| West bounding | 90.0 |
| East bounding | 175.0 |
| Date of metadata | 10th July 2018 |

| **Habitat Rugosity** | |
|---|---|
| Description | This map presents a Vector Ruggedness Measure (VRM) of benthic terrain as a proxy of benthic habitat heterogeneity. The VRM index ranged from 0.1 (areas with low terrain variations to 0.9 (areas with high terrain variations). |
| Temporal range | Refers to the native sources |
| Geographical range | Coral Triangle of the Indo Pacific Realm |
| Citations | Asaad, I., Lundquist, C. J., Erdmann, M. V., Van Hooidonk, R., & Costello, M. J. (2018). Designating spatial priorities for marine biodiversity conservation in the Coral Triangle. *Frontiers in Marine Science, 5*, 400. doi: 10.3389/fmars.2018.00400 |
| Original datasets / citations | Basher, Z., Bowden, D.A., & Costello, M.J. (2014). Global Marine Environment Datasets (GMED)-World Wide Web electronic publication. Version 1.0 (Rev.01.2014). Retrieved 01 June 2016 http://gmed.auckland.ac.nz

Wright, D., Pendleton, M., Boulware, J., Walbridge, S., Gerlt, B., Eslinger, D., Sampson, D., & Huntley, E. (2012). ArcGIS Benthic Terrain Modeler (BTM), v. 3.0, Environmental Systems Research Institute (ESRI), NOAA Coastal Services Center, Massachusetts Office of Coastal Zone Management. Redland - CA. |
| Purpose of creation | To identify areas of biodiversity importance within the Coral Triangle based on the habitat heterogeneity. |
| Creation methodology | The dataset of a Vector Ruggedness Measure (VRM) of benthic terrain was analyzed to measure benthic terrain rugosity and topographic ruggedness as an indicator of benthic habitat heterogeneity. To quantify this index, bathymetry data were extracted from GMED (Global Marine Environment Datasets) (Basher *et al.*, 2014) and analyzed it using the Benthic Terrain Modeller (BTM) 3.0 of ArcGIS 10.5 (Wright *et al.*, 2012). The VRM index ranged from 0.1 to 0.9, and were classified into 10 equal interval classes.
The methodology is fully described at Asaad *et al* (2018b) |
| Version | 1 (July 2018) |
| Keywords: | Coral Triangle, biodiversity importance, biodiversity feature, habitat rugosity |
| Category | Biodiversity Features |

| | |
|---|---|
| Limitations: | This dataset used bathymetry and slope data to generate benthic terrain rugosity and topographic ruggedness as a proxy of habitat heterogeneity. However, bathymetry and slope are not the only drivers of habitat heterogeneity in several habitats such as soft sediment habitats. |
| Main access/use constraint: | Creative Commons Attribution 4.0 (CC BY 4.0) |
| Contact Organization | Institute of Marine Science, University of Auckland |
| Name | Irawan Asaad |
| City | Auckland |
| Country | New Zealand |
| email | i.asaad@auckland.ac.nz |
| Data format | Geodatabase (Grid square cells; polygon) |
| Distribution format | GeoJSON |
| Dataset size | 2.27 MB |
| Webpage | www.marine.auckland.ac.nz/CT_Biodiversity |
| Otherweb page | https://uoa.maps.arcgis.com/apps/webappviewer/index.html?id=1406b9131245493195c12a1df3d2ada6 |
| Resolution / scale | 50 km |
| Reference System | WGS 84 |
| North bounding | 22.0 |
| South bounding | -16.0 |
| West bounding | 90.0 |
| East bounding | 175.0 |
| Date of metadata | 10th July 2018 |

| **Anthropogenic Pressure** | |
|---|---|
| Description | This map presents a spatial distribution of anthropogenic pressure to marine environments. This map was generated based on the cumulative impact of 19 different types of anthropogenic stressors developed by by Halpern *et al.* (2008;2015). The anthropogenic pressure value ranged from 0 – 15.4, indicating areas from low to high human-induced pressure. |
| Temporal range | Refers to the native sources |
| Geographical range | Coral Triangle of the Indo Pacific Realm |
| Citations | Asaad, I., Lundquist, C. J., Erdmann, M. V., Van Hooidonk, R., & Costello, M. J. (2018). Designating spatial priorities for marine biodiversity conservation in the Coral Triangle. *Frontiers in Marine Science, 5*, 400. doi: 10.3389/fmars.2018.00400 |
| Original datasets / citations | Halpern, B.S., Walbridge, S., Selkoe, K.A., Kappel, C.V., Micheli, F., D'Agrosa, C., Bruno, J.F., Casey, K.S., Ebert, C., Fox, H. E., Fujita, R., Heinemann, D., Lenihan, H.S., Madin, E.M.P., Perry, M.T., Selig, E.R., Spalding, M., Steneck, R., & Watson, R. (2008). A global map of human impact on marine ecosystems. *Science, 319*(5865), 948-952. DOI: 10.1126/science.1149345.
Halpern, B.S., Frazier, M., Potapenko, J., Casey, K.S., Koenig, K., Longo, C., Lowndes, J.S., Rockwood, R.C., Selig, E.R., & Selkoe, K.A. (2015a). Spatial and temporal changes in cumulative human impacts on the world's ocean. *Nature communications*, 6. |

| | |
|---|---|
| | Halpern, B. Frazier, M., Potapenko, J., Casey, K.S., Koenig, K., Longo, C., Lowndes, J.S., Rockwood, R.C., Selig, E.R., & Selkoe, K.A. (2015b). Cumulative human impacts: raw stressor data (2008 and 2013). Accessed 01/06/2016. https://knb.ecoinformatics.org/. |
| Purpose of creation | To identify the spatial distribution of anthropogenic pressure within the Coral Triangle. |
| Creation methodology | The spatial distribution of anthropogenic pressure to marine environments was retrieved from the database of cumulative human impacts on the world's oceans developed by Halpern *et al.* (Halpern *et al.*, 2008; Halpern *et al.*, 2015a; 2015b) This dataset was based on the cumulative impact of 19 different types of anthropogenic stressors: land-based drivers (nutrient inputs, organic and inorganic pollution, and population density), ocean-based drivers (commercial fishing, artisanal fishing, benthic structures, shipping lanes, invasive species, and pollution), and climate change (sea-level rise, sea-surface temperature anomalies, ultraviolet radiation and acidification). The methodology is fully described at Asaad *et al* (2018b)

With spatial resolution of 5 km, the anthropogenic pressure value of each cell was ranged from 0 – 15.4. Thus, the cells were classified into 10 equal interval classes. |
| Version | 1 (July 2018) |
| Keywords: | Coral Triangle, biodiversity importance, biodiversity feature, habitat rugosity |
| Category | Biodiversity Features |
| Limitations: | This dataset developed based on the cumulative human impact at global scale. The dataset captured trends and variations of pressure at local scale, but has a limitation to identify local events (e.g. impact of dynamite and poisonous fishing). |
| Main access/use constraint: | Creative Commons Attribution 4.0 (CC BY 4.0) |
| Contact Organization | Institute of Marine Science, University of Auckland |
| Name | Irawan Asaad |
| City | Auckland |
| Country | New Zealand |
| email | i.asaad@auckland.ac.nz |
| Data format | Geodatabase (Grid square cells; polygon) |
| Distribution format | GeoJSON |
| Dataset size | 0.76 MB |
| Webpage | *www.marine.auckland.ac.nz/CT_Biodiversity* |
| Otherweb page | https://uoa.maps.arcgis.com/apps/webappviewer/index.html?id=1406b9131245493195c12a1df3d2ada6 |
| Resolution / scale | 5 km |
| Reference System | WGS 84 |
| North bounding | 22.0 |
| South bounding | -16.0 |
| West bounding | 90.0 |
| East bounding | 175.0 |
| Date of metadata | 10th July 2018 |

| **Climate change Pressure** | |
|---|---|
| Description | This map presents a spatial distribution of sea surface thermal stress level. This map was generated based on the average of Degree Heating Weeks (DHW) datasets developed by Van Hooidonk *et al* (2016). The projected thermal stress index ranged from 5.6 – 20.2, indicating areas from low to high vulnerability to climate change. |
| Temporal range | Refers to the native sources |
| Geographical range | Coral Triangle of the Indo Pacific Realm |
| Citations | Asaad, I., Lundquist, C. J., Erdmann, M. V., Van Hooidonk, R., & Costello, M. J. (2018). Designating spatial priorities for marine biodiversity conservation in the Coral Triangle. *Frontiers in Marine Science, 5*, 400. doi: 10.3389/fmars.2018.00400 |
| Original datasets / citations | Van Hooidonk, R., Maynard, J., Tamelander, J., Gove, J., Ahmadia, G., Raymundo, L., Williams, G., Heron, S.F., & Planes, S. (2016). Local-scale projections of coral reef futures and implications of the Paris Agreement. *Scientific reports, 6*, 39666. |
| Purpose of creation | To identify the spatial distribution of sea surface thermal stress level pressure as an indicator of climate - induced stressor within the Coral Triangle. |
| Creation methodology | The dataset of the sea-surface thermal stress level was derived from Van Hooidonk *et al.* (2016). This dataset was based on the average of projected Degree Heating Weeks (DHW) (2006 to 2099) under RCP8.5 scenario. Degree heating weeks (DHW) is a measurement to assess patterns of sea surface temperature (SST) variability by combining the intensity and duration of thermal stress in order to predict coral bleaching (Liu *et al.*, 2003). With spatial resolution of 5 km, the thermal stress value of each cell was ranged from 5.6 – 20.2. Thus, the cells were classified into 10 equal interval classes. The methodology is fully described at Asaad *et al* (2018b) |
| Version | 1 (July 2018) |
| Keywords: | Coral Triangle, biodiversity importance, biodiversity feature, climate-induced pressure |
| Category | Biodiversity Features |
| Limitations: | |
| Main access/use constraint: | Creative Commons Attribution 4.0 (CC BY 4.0) |
| Contact Organization | Institute of Marine Science, University of Auckland |
| Name | Irawan Asaad |
| City | Auckland |
| Country | New Zealand |
| email | i.asaad@auckland.ac.nz |
| Data format | Geodatabase (Grid cells; polygon) |
| Distribution format | GeoJSON |
| Dataset size | 1.02 MB |
| Webpage | *www.marine.auckland.ac.nz/CT_Biodiversity* |
| Otherweb page | http://uoa.maps.arcgis.com/apps/webappviewer/index.html?id=1406b9131245493195c12a1df3d2ada6 |
| Resolution / scale | 5 km |
| Reference System | WGS 84 |
| North bounding | 22.0 |

| South bounding | -16.0 |
|---|---|
| West bounding | 90.0 |
| East bounding | 175.0 |
| Date of metadata | 10th July 2018 |

| **Regional biodiversity hotspots** | |
|---|---|
| Description | This map presents clusters of areas of biodiversity importance within the Coral Triangle. Retrieved from datasets of areas of biodiversity importance developed by Asaad *et al*., (2018a). The regional biodiversity hotspots were classified into 3 classes of biodiversity hotspots (high, medium and low) and 1 class not significant. |
| Temporal range | Refers to the native sources |
| Geographical range | Coral Triangle of the Indo Pacific Realm |
| Citations | Asaad, I., Lundquist, C. J., Erdmann, M. V., & Costello, M. J. (2018). Delineating priority areas for marine biodiversity conservation in the Coral Triangle. *Biological Conservation, 222*, 198-211. doi: dx.doi.org/10.1016/j.biocon.2018.03.037 |
| Original datasets / citations | Asaad, I., Lundquist, C. J., Erdmann, M. V., & Costello, M. J. (2018). Delineating priority areas for marine biodiversity conservation in the Coral Triangle. *Biological Conservation, 222*, 198-211. doi: dx.doi.org/10.1016/j.biocon.2018.03.037 |
| Purpose of creation | To identify "clustered hotspots" (*i.e.,* groups of cells) of biodiversity significance within the Coral Triangle. |
| Creation methodology | To evaluate clustered areas of biodiversity importance, Asaad et al (2018a) used multi-criteria analysis to five ecological criteria (sensitive habitat, species richness, the presence of species of conservation concern, the occurrence of restricted-range species, areas of importance for particular life history stages). Areas of biodiversity importance were identified by superimposing each of the different criterion.

Using a grid approach of half-degree cells (0.5°), the regional-level analyses were conducted by evaluating clustered areas of biodiversity importance using the hotspots analysis tool in ArcGIS 10.5. The hotspot tool identifies the spatial patterns of data based on the Getis-Ord GI* statistics, clustered the cells from hotspot (high score cells) to coldspots (low score cells).

The methodology is fully described at Asaad *et al* (2018a). |
| Version | 1 (July 2018) |
| Keywords: | Coral Triangle, biodiversity importance, biodiversity feature, biodiversity hotspots |
| Category | Biodiversity Features |
| Limitations: | To have a comprehensive assessment of the biodiversity conservation value of the region, other ecological criteria are recommended: unique and rare habitats, representativeness and ecological integrity. In addition, in the absence of deep-sea biodiversity datasets, the areas of biodiversity importance may exhibit geographical bias toward shallow-water area. |
| Main access/use constraint: | Creative Commons Attribution 4.0 (CC BY 4.0) |
| Contact Organization | Institute of Marine Science, University of Auckland |
| Name | Irawan Asaad |
| City | Auckland |
| Country | New Zealand |
| email | i.asaad@auckland.ac.nz |

| | |
|---|---|
| Data format | Geodatabase (Grid cells; polygon) |
| Distribution format | GeoJSON |
| Dataset size | 1.82 MB |
| Webpage | *www.marine.auckland.ac.nz/CT_Priority* |
| Otherweb page | http://uoa.maps.arcgis.com/apps/webappviewer/index.html?id=429a21089ce243eb9d683b23d7c53da2 |
| Resolution/scale | 55 km |
| Reference System | WGS 84 |
| North bounding | 22.0 |
| South bounding | -16.0 |
| West bounding | 90.0 |
| East bounding | 175.0 |
| Date of metadata | 10th July 2018 |

| Sites of biodiversity importance | |
|---|---|
| Description | This map shows distribution of sites of biodiversity importance within the Coral Triangle. Retrieved from datasets of areas of biodiversity importance developed by Asaad *et al*., (2018a). The site based biodiversity importance were classified into 5 classes ((high, medium-high, medium, medium-low and low). |
| Temporal range | Refers to the native sources |
| Geographical range | Coral Triangle of the Indo Pacific Realm |
| Citations | Asaad, I., Lundquist, C. J., Erdmann, M. V., & Costello, M. J. (2018). Delineating priority areas for marine biodiversity conservation in the Coral Triangle. *Biological Conservation, 222*, 198-211. doi: dx.doi.org/10.1016/j.biocon.2018.03.037 |
| Original datasets / citations | Asaad, I., Lundquist, C. J., Erdmann, M. V., & Costello, M. J. (2018). Delineating priority areas for marine biodiversity conservation in the Coral Triangle. *Biological Conservation, 222*, 198-211. doi: dx.doi.org/10.1016/j.biocon.2018.03.037 |
| Purpose of creation | To identify sites of biodiversity "clustered hotspots" (*i.e.,* groups of cells) of biodiversity significance within the Coral Triangle. |
| Creation methodology | To identify sites of biodiversity importance, Asaad et al (2018a) used multi-criteria analysis to five ecological criteria (sensitive habitat, species richness, the presence of species of conservation concern, the occurrence of restricted-range species, areas of importance for particular life history stages). Areas of biodiversity importance were identified by superimposing each of the different criterion. Using a grid approach of half-degree cells (0.5°), the site-based analysis identifies specific sites of highest biodiversity importance by analyzing the biodiversity score of each cell. The higher the score, the higher their biodiversity importance.

The methodology is fully described at Asaad *et al* (2018a) |
| Version | 1 (July 2018) |
| Keywords: | Coral Triangle, biodiversity importance, biodiversity feature, biodiversity hotspots |
| Category | Biodiversity Features |
| Limitations: | To have a comprehensive assessment of the biodiversity conservation value of the region, other ecological criteria are recommended: unique and rare habitats, representativeness and ecological integrity. In addition, in the absence of deep-sea biodiversity datasets, the areas of biodiversity importance may exhibit geographical bias toward shallow-water area. |
| Main access/use constraint: | Creative Commons Attribution 4.0 (CC BY 4.0) |

| | |
|---|---|
| Contact Organization | Institute of Marine Science, University of Auckland |
| Name | Irawan Asaad |
| City | Auckland |
| Country | New Zealand |
| email | i.asaad@auckland.ac.nz |
| Data format | Geodatabase (Grid cells; polygon) |
| Distribution format | GeoJSON |
| Dataset size | 0.12 MB |
| Webpage | www.marine.auckland.ac.nz/CT_Priority |
| Otherweb page | http://uoa.maps.arcgis.com/apps/webappviewer/index.html?id=429a21089ce243eb9d683b23d7c53da2 |
| Resolution / scale | 55 km |
| Reference System | WGS 84 |
| North bounding | 22.0 |
| South bounding | -16.0 |
| West bounding | 90.0 |
| East bounding | 175.0 |
| Date of metadata | 10th July 2018 |

| **Marine Protected Area (MPA) Network Expansion: Regional priority areas** | |
|---|---|
| Description | This map presents spatial distribution of regional priority areas with three expansion scenario layers (*e.g.,* expansion of the MPA network from existing coverage to 10%, 20% and 30 % of the Economic Exclusive Zone (EEZ) area). Retrieved from datasets of Coral Triangle Marine Protected Area (MPA) System Expansion developed by Asaad *et al*., (2018b). |
| Temporal range | Refers to the native sources |
| Geographical range | Coral Triangle of the Indo Pacific Realm |
| Citations | Asaad, I., Lundquist, C. J., Erdmann, M. V., Van Hooidonk, R., & Costello, M. J. (2018). Designating spatial priorities for marine biodiversity conservation in the Coral Triangle. *Frontiers in Marine Science, 5*, 400. doi: 10.3389/fmars.2018.00400 |
| Original datasets / citations | Asaad, I., Lundquist, C. J., Erdmann, M. V., Van Hooidonk, R., & Costello, M. J. (2018). Designating spatial priorities for marine biodiversity conservation in the Coral Triangle. *Frontiers in Marine Science, 5*, 400. doi: 10.3389/fmars.2018.00400 |
| Purpose of creation | To develop a prioritization scenario for expansion of the MPA system in the Coral Triangle and provide a conservation strategy to expand the CT MPA system to fulfill the obligations to the CBD-Aichi Biodiversity Target 11, and to achieve Goal 14 of the UN-United Nations-Sustainable Development Goals |
| Creation methodology | To guide the identification of an effective MPA system, Asaad et al (2018b) conducted prioritization analyses using systematic conservation planning software of *Zonation*. The prioritization scenarios were based on seven sets of biodiversity features (biogenic habitat, habitat rugosity, species richness, distribution of threatened and endemic species, areas important for sea turtle); two types of threat (anthropogenic and climate change induced pressure); and the coverage of the existing MPA network.

Analysis were conducted by compared changes in the proportion of biodiversity features protected and the spatial distribution of priority areas with increasing proportions of the CT region placed into an MPA network. That is, it projected the expansion of the MPA system in the Coral Triangle from the present 1.8% to 10%, 20% and 30% of the combined EEZ area. Using a grid approach of 0.5 km resolution, Regional analyses were performed for the full CT EEZ region

The methodology is fully described at Asaad *et al* (2018b). |

| | |
|---|---|
| Version | 1 (July 2018) |
| Keywords: | Coral Triangle, biodiversity importance, biodiversity feature, biodiversity hotspots |
| Category | Biodiversity Features |
| Limitations: | |
| Main access/use constraint: | Creative Commons Attribution 4.0 (CC BY 4.0) |
| Contact Organization | Institute of Marine Science, University of Auckland |
| Name | Irawan Asaad |
| City | Auckland |
| Country | New Zealand |
| email | i.asaad@auckland.ac.nz |
| Data format | Geodatabase (Grid cells; polygon) |
| Distribution format | GeoJSON |
| Dataset size | 14.65 MB |
| Webpage | www.marine.auckland.ac.nz/CT_MPA |
| Otherweb page | http://uoa.maps.arcgis.com/apps/webappviewer/index.html?id=2f36a9ec18674a13a4e57fd290fc020a |
| Resolution / scale | 0.5 km |
| Reference System | WGS 84 |
| North bounding | 22.0 |
| South bounding | -16.0 |
| West bounding | 90.0 |
| East bounding | 175.0 |
| Date of metadata | 10th July 2018 |

| Marine Protected Area (MPA) Network Expansion: National Priority Areas | |
|---|---|
| Description | This map presents spatial distribution of national priority areas with six layers of scenarios representing national MPA network expansion for Indonesia, Malaysia, the Philippines, Papua New Guinea, Solomon Islands and Timor Leste. Each country has three expansion scenario layers (*e.g.,* expansion of the MPA network from existing coverage to 10%, 20% and 30 % of the Economic Exclusive Zone (EEZ) area). Retrieved from datasets of Coral Triangle Marine Protected Area (MPA) System Expansion developed by Asaad *et al*., (2018b). |
| Temporal range | Refers to the native sources |
| Geographical range | Coral Triangle of the Indo Pacific Realm |
| Citations | Asaad, I., Lundquist, C. J., Erdmann, M. V., Van Hooidonk, R., & Costello, M. J. (2018). Designating spatial priorities for marine biodiversity conservation in the Coral Triangle. *Frontiers in Marine Science, 5*, 400. doi: 10.3389/fmars.2018.00400 |
| Original datasets / citations | Asaad, I., Lundquist, C. J., Erdmann, M. V., Van Hooidonk, R., & Costello, M. J. (2018). Designating spatial priorities for marine biodiversity conservation in the Coral Triangle. *Frontiers in Marine Science, 5*, 400. doi: 10.3389/fmars.2018.00400 |
| Purpose of creation | To develop a prioritization scenario of each country in the Coral Triangle to expand their MPA system and provide a conservation strategy to fulfill the obligations to the CBD-Aichi Biodiversity Target 11, and to achieve Goal 14 of the UN-United Nations-Sustainable Development Goals |
| Creation methodology | To guide the identification of an effective MPA system in each CT country, Asaad *et al* (2018b) conducted prioritization analyses using systematic conservation planning software of *Zonation.* The prioritization scenarios were based on seven sets of biodiversity features (biogenic habitat, habitat rugosity, species richness, distribution of threatened and endemic species, areas important for sea turtle); two types of threat (anthropogenic and climate change induced pressure); and the coverage of the existing MPA network. |

| | Analysis were conducted by compared changes in the proportion of biodiversity features protected and the spatial distribution of priority areas with increasing proportions of the CT region placed into an MPA network. That is, it projected the expansion of the MPA system in the Coral Triangle from the present 1.8% to 10%, 20% and 30% of the combined EEZ area. Using a grid approach of 0.5 km resolution, national analyses were performed individually on each CT country national EEZ.

The methodology is fully described at Asaad *et al* (2018b). |
|---|---|
| Version | 1 (July 2018) |
| Keywords: | Coral Triangle, biodiversity importance, biodiversity feature, biodiversity hotspots |
| Category | Biodiversity Features |
| Limitations: | |
| Main access/use constraint: | Creative Commons Attribution 4.0 (CC BY 4.0) |
| Contact Organization | Institute of Marine Science, University of Auckland |
| Name | Irawan Asaad |
| City | Auckland |
| Country | New Zealand |
| email | i.asaad@auckland.ac.nz |
| Data format | Geodatabase (Grid cells; polygon) |
| Distribution format | GeoJSON |
| Dataset size | 4.6 MB |
| Webpage | *www.marine.auckland.ac.nz/CT_MPA* |
| Otherweb page | http://uoa.maps.arcgis.com/apps/webappviewer/index.html?id=2f36a9ec18674a13a4e57fd290fc020a |
| Resolution / scale | 0.5 km |
| Reference System | WGS 84 |
| North bounding | 22.0 |
| South bounding | -16.0 |
| West bounding | 90.0 |
| East bounding | 175.0 |
| Date of metadata | 10th July 2018 |

| **Marine protected areas (MPA) coverage** | |
|---|---|
| Description | This map presents spatial distribution of marine protected areas within the Coral Triangle. This dataset consisted of 678 MPA boundaries, retrieved from the World Database of Protected Areas-WDPA (www.protectedplanet.net), the Coral Triangle Atlas (ctatlas.reefbase.org) and the Indonesian database of marine protected areas. |
| Temporal range | Refers to the native sources |
| Geographical range | Coral Triangle of the Indo Pacific Realm |
| Citations | Asaad, I., Lundquist, C. J., Erdmann, M. V., & Costello, M. J. (2018). Delineating priority areas for marine biodiversity conservation in the Coral Triangle. *Biological Conservation, 222*, 198-211. doi: dx.doi.org/10.1016/j.biocon.2018.03.037 |
| Original datasets / citations | Cros, A., Fatan, N.A., White, A., Teoh, S.J., Tan, S., Handayani, C., Huang, C., Peterson, N., Li, R.V., Siry, H.Y., Fitriana, R., Gove, J., Acoba, T., Knight, M., Acosta, R., Andrew, N., & Beare, D. (2014a). The Coral Triangle Atlas: An Integrated Online Spatial Database System for Improving Coral Reef Management. *Plos One, 9*(6). DOI: 10.1371/journal.pone.0096332. http://ctatlas.reefbase.org/
IUCN & UNEP-WCMC. (2016). The World Database on Protected Areas (WDPA). Accessed 01/08/2016, from UNEP - World Conservation Monitoring Centre. www.protectedplanet.net. Cambridge-UK
MoF-MoMAF. (2010). Ecological representation gap analysis for conservation areas in Indonesia (pp. 29). |

| | |
|---|---|
| | Ministry of Forestry and Ministry of Marine Affairs and Fisheries. Jakarta-Indonesia. MoMAF. (2016a). Database of Marine Protected Areas in Indonesia. Ministry of Marine Affairs and Fisheries. Jakarta - Indonesia. |
| Purpose of creation | To identify areas of biodiversity importance and to develop a geographic prioritization of marine biodiversity conservation in the Coral Triangle region. |
| Creation methodology | This dataset is a combined data of 3 sources: the World Database of Protected Areas-WDPA, the Coral Triangle Atlas and the Indonesian database of marine protected areas. In total, there are more than 2000 MPA exists in the region, but this dataset contains only 678 MPA boundaries in a polygon format. We excluded MPAs which had missing boundaries or were represented only by point locations (longitude and latitude coordinates) as they may reduce the validity.

 The layers' attribute table provides detailed information following its native sources (WDPA, CTAtlas) (e.g., information of Name, Local Name, Designation Type, IUCN Category, coverage etc.) (IUCN & UNEP-WCMC,2016; Cros *et al.*,2014) with amendment and adjustment from local sources (Indonesian database).
 To allow simple indexing, a new CT MPAs ID format (MPA_ID) is introduced. The new ID consists of 10 digits: " C IC XXXX yyy "

 Where:
 C = Country; 1 = Indonesia, 2 = Malaysia,  3 = Philippines, 4 = Papua New Guinea,
        5 = Solomon Islands, and 6 = Timor Leste
 IC = IUCN MPAs Category; Strict Nature Reserve (1a = 11, 1b = 12), National Park (20),
        Habitat and Species Management Areas (40), Protected Landscape/Seascape (50) and
        Managed Resources Protected Areas (60)
 XXXX = Establishment year (*e.g.,* 1980)
 yyy = Number; ordered based on their establishment year

 The methodology is fully described at Asaad *et al* (2018b). |
| Version | 1 (July 2018) |
| Keywords: | Coral Triangle, biodiversity importance, biodiversity feature, biodiversity hotspots |
| Category | Biodiversity Features |
| Limitations: | The total coverage of MPA summed over the available polygon boundaries (240,443 km$^2$) is larger than the total coverage of MPA officially reported by the CT countries (200,881 km$^2$) (White *et al.*, 2014).  The discrepancy in MPA coverage occurs as some protected areas have both terrestrial and marine components (*e.g.,* coastline, beaches or small islands), and there were inconsistencies between the official documents and the accompanying GIS spatial boundary datasets. |
| Main access/use constraint: | Creative Commons Attribution 4.0 (CC BY 4.0) |
| Contact Organization | Institute of Marine Science, University of Auckland |
| Name | Irawan Asaad |
| City | Auckland |
| Country | New Zealand |
| email | i.asaad@auckland.ac.nz |
| Data format | Geodatabase (Polygon) |
| Distribution format | GeoJSON |
| Dataset size | 4.1 MB |
| Webpage | *www.marine.auckland.ac.nz/CT_Biodiversity* |
| Otherweb page | https://uoa.maps.arcgis.com/apps/webappviewer/index.html?id=1406b9131245493195c12a1df3d2ada6 |
| Resolution / scale | 0.5 km |

| | |
|---|---|
| Reference System | WGS 84 |
| North bounding | 22.0 |
| South  bounding | -16.0 |
| West bounding | 90.0 |
| East bounding | 175.0 |
| Date of metadata | 10th  July 2018 |

---

## Author Comment (AC3) · 5 Nov 2018

**1. Referee comments**

The Introduction section provides a good generic introduction to online databases and web maps but would benefit from a little more detail on the threats and conservation efforts in the Coral Triangle region that need better spatial data. Who exactly would benefit and why? Outline specific examples where this online map product will enhance decision making in the region? For example, in Line 189-192 of the Discussion you mention the importance of spatial prioritization and 'enable an efficient decision making process'. It would be good if you introduce these projects and processes in the Introduction.

[Figure]

2. Response

We have amended the Introduction and have included the background, conservation efforts, the importance and who are the beneficiaries of this atlas.

3. It is now read:

"To take advantage of the potential of web-mapping, here we developed a web-mapping application for the Coral Triangle (CT) region of the Indo Pacific realm, a global hotspot for marine biodiversity conservation due to its superlative species richness and endemicity (Hoeksema, 2007; Allen, 2008; Veron et al., 2009; Polidoro et al., 2010; Walton et al., 2014; Saeedi et al., 2016). Because the region has the highest density or marine species of anywhere in the ocean, it is a priority for marine conservation. Furthermore, a large amount of biodiversity and natural resources data have been collected for decades by scientists and numerous conservation programmes. However, data repositories are scattered, and access to such data are limited. Previously, a systematic geographic prioritization to develop Marine Protected Area (MPA) system was conducted (Asaad. et al., 2018a; 2018b), but this alone does not make the information easily available to the public. In addition, previous online atlas of the CT was developed to support coral reef management and provided biophysical and MPA data from the region (Cros et al., 2014), however an updated, more systematic and comprehensive "biodiversity informatics" datasets are required to showcase all of the available data in the region. Further, this web-atlas is aimed to support the objective of the CTI-CFF initiative (the Coral Triangle Initiative on Coral Reefs, Fisheries and Food Security). The CTI-CFF initiative is a multilateral partnership of six countries (Indonesia, Malaysia, Papua New Guinea, the Philippines, Solomon Islands, and Timor-Leste) to working collaboratively to conserve and sustainably manage their coastal and marine resources (CTI-CFF, 2009, 2013). One of the objectives of this initiative is to establish and effectively manage Marine Protected Areas (MPA) network, which emphasizes the importance of developing and managing MPA throughout the region. In addition, an MPA system framework was developed to guide the development of network of MPAs

in the region (Walton et al., 2014). Thus, the collections of geospatial data collated on this online GIS database are designed to support and assist in the development of marine protected areas and management of marine resources in the region. By giving an access to policymakers, scientific communities, and the general public to the most comprehensive, up-to-date and integrated spatial information available for the Coral Triangle"

1. Referee comments

Table 1 was repeated again on pages 19-22 in my version of the PDF.

2. Response

We have remove the table of dataset specification in the appendices

1. Referee comments

- Line 60-61 - typo

- Line 75 - typo with tense 'develop' should be 'developed' and 'geo-reference' should be 'geo-referenced'

- Line 79 - typo 'and and'

- Line 84 -ArcGIS Pro 2.0 was instead of 'The ArcGIS Pro 2.0 were

- Line 85 - and design three instead of 'designed these three'

- Line 86 - was used instead of 'were used'

- Line87 - computer or other electronic device connected to (no need for plurals)

- Line 88- hosted by ArcGIS Online not 'by the ArcGIS Online'

2. Response

We have corrected all of the typos

1. Referee comments

Line 95 - refer to Table 1 at the end of each bullet point where appropriate

2.. Response

We have added the citation (Table 1).

1. Referee comments

- Line 96-97 - Briefly mention how these layers were defined and provide citations

2. Response

We have provided the brief info of each layers and citation in the Table 1.

1. Referee comments

Line 98 - Briefly mention how these layers were defined and provide citations

Response

We have provided the brief info of each layers and citation in the Table 1.

1. Referee comments

Line 106 - The Table 1 has been referred to as 5.1 and occasionally a chapter has been referred to so please revise throughout to be consistent for this manuscript. Same for your figure numbers. It sounds like this manuscript was originally written as a chapter

Line 110 - see comment above

2.Response

We have corrected all of the table and figure number

1. Referee comments Line 110-111 - provide URL for documentation if available online

2. Response

We have provided the URL of original sources in the reference sections, and also provided a metadata of each layer. For your considerations, we have attached the metadata in the last section of this reviewer responses.

1. Referee comments

Table 2 - I don't think you need to show the widgets. These are generic to ESRI app developer and do not provide useful information. It is something that you expect to see in a user manual.

2. Response

We slightly disagree. Although the provided widgets are generic to ESRI apps, but we opted to show the widgets so users (particularly the non-GIS users) may explore the functionality of the atlas based on the provided widgets.

1. Referee comments

In your description of methods there is no mention of uncertainty or error in your datasets. The metadata should include information to allow the user to assess uncertainty. You should also provide discussion on this issue and on any perceived scale (temporal and spatial) limitations of the data in this manuscript

2. Response

We have provided the metadata and describe the uncertainty and limitation of our data

1. Referee comments

- Line 188 - to develop geospatial tools to support instead of 'and develop a geospatial tool'

- Line 206 - typo - remove 'and' after 'provided'

- Line 213 - typo - remove 'areas' before 'priority'

- Line 217 - remove 'this' before 'digital maps'

2. Response

We have corrected the typos

1. Referee comments

229-238 - The conclusion is basically just a repetition of the same information in the manuscript. It would be more useful for you to think about future applications (e.g., modelling biological distributions, predicting spatial change, mapping vulnerability to threats, spatial resilience) and improvements to tool functionality.

2. Response

We have amended the conclusions.

3. It is now read:

"To take advantage of the potential of web-mapping, here we developed a web-mapping application for the Coral Triangle (CT) region of the Indo Pacific realm, a global hotspot for marine biodiversity conservation due to its superlative species richness and endemicity (Hoeksema, 2007; Allen, 2008; Veron et al., 2009; Polidoro et al., 2010; Walton et al., 2014; Saeedi et al., 2016). Because the region has the highest density or marine species of anywhere in the ocean, it is a priority for marine conservation. Furthermore, a large amount of biodiversity and natural resources data have been collected for decades by scientists and numerous conservation programmes. However, data repositories are scattered, and access to such data are limited. Previously, a systematic geographic prioritization to develop Marine Protected Area (MPA) system was conducted (Asaad. et al., 2018a; 2018b), but this alone does not make the information easily available to the public. In addition, previous online atlas of the CT was developed to support coral reef management and provided biophysical and MPA data from the region (Cros et al., 2014), however an updated, more systematic and comprehensive "biodiversity informatics" datasets are required to showcase all of the available data in the region. Further, this webatlas is aimed to support the objective of the CTI-CFF initiative (the Coral Triangle Initiative on Coral Reefs, Fisheries and Food Security). The CTI-CFF initiative is a multilateral partnership of six countries (Indonesia, Malaysia, Papua New Guinea, the Philippines, Solomon Islands, and Timor-Leste) to working collaboratively to conserve and sustainably manage their coastal and marine resources (CTI-CFF, 2009, 2013). One of the objectives of this initiative is to establish and effectively manage Marine Protected Areas (MPA) network, which emphasizes the importance of developing and managing MPA throughout the region. In addition, an MPA system framework was developed to guide the development of network of MPAs in the region (Walton et al., 2014). Thus, the collections of geospatial data collated on this online GIS database are designed to support and assist in the development of marine protected areas and management of marine resources in the region. By giving an access to policymakers, scientific communities, and the general public to the most comprehensive, up-to-date and integrated spatial information available for the Coral Triangle"

Please also note the supplement to this comment:
https://www.earth-syst-sci-data-discuss.net/essd-2018-80/essd-2018-80-AC3-supplement.pdf

---

## Author Comment (AC4) · 5 Nov 2018

Dear Prof. Huettmann

We would like to thank you for your efforts in coordinating the review process. We have responded to all of the comments provided by the referees and revised our manuscript accordingly. The referees provided a large range of comments that have improved our manuscript.

Following your suggestion and the referees comments, we have revised the title to " An interactive atlas for marine biodiversity conservation in the Coral Triangle", as we think that "an Atlas" is more suitable to explain our data-sets compendium than a simple "Digital Map". We also have amended the introduction, discussion sections and

develop a metadata to our dataset.

Again, thank you for support.

Regards,

Irawan Asaad